# Heterochromatic gene silencing controls CD4+ T cell susceptibility to regulatory T cell-mediated suppression in a murine allograft model

Julie Noguerol [1], Karl Laviolette[1,6], Margot Zahm [1,6], Adeline Chaubet [1], Ambrine Sahal[2], Claire Détraves[1], Romain Torres[1], Clothilde Demont[1], Véronique Adoue[1], Carine Joffre [2], Florence Cammas [3,4,5], Joost PM van Meerwijk [1] & Olivier P. Joffre [1] ✉

Protective immune responses require close interactions between conventional (Tconv) and regulatory T cells (Treg). The extracellular mediators and signaling events that regulate the crosstalk between these CD4+ T cell subsets have been extensively characterized. However, how Tconv translate Treg-dependent suppressive signals at the chromatin level remains largely unknown. Here we show, using a murine bone marrow allograft model in which graft rejection is coordinated by CD4+ T cells and can be inhibited by Treg, that Treg-mediated T cell suppression involves Heterochromatin Protein 1 α (HP1α)-dependent gene silencing. Unexpectedly, our screen also reveals that T cells deficient for HP1γ or the methyltransferase SUV39H1 are better repressed by Treg than their wild-type counterparts. Mechanistically, our transcriptional and epigenetic profiling identifies HP1γ as a negative regulator of a gene network functionally associated with T-cell exhaustion, including those encoding the inhibitory receptors PD-1 and LAG-3. In conclusion, we identify HP1 variants as rheostats that finely tune the balance between tolerance and immunity. While HP1α converts immunosuppressive signals into heterochromatin-dependent gene silencing mechanisms, HP1γ adjusts Tconv sensitivity to inhibitory environmental signals.

The immune system is highly efficient at protecting the host against a wide diversity of endogenous and exogenous dangers. This efficiency derives to a large extent from the ability of CD4+ T cells to adapt their phenotype and function to the nature of the threat. Upon priming by dendritic cells, and depending on the nature of the pathological stimulus, naive CD4+ T cells are exposed to distinct molecular environments. In response, they mobilize different gene networks that establish lineage-specific developmental programs, and coordinate the acquisition of specific phenotype and functions. Accordingly, CD4+ T cells can differentiate into a large variety of functionally distinct T

[1]Infinity, Toulouse Institute for Infectious and Inflammatory Diseases, University of Toulouse, Inserm U1291, CNRS U5051 Toulouse, France. [2]Centre de Recherche en Cancérologie de Toulouse, Université de Toulouse, Inserm U1037, CNRS U5071 Toulouse, France. [3]Institut de Recherche en Cancérologie de Montpellier, INSERM U1194, Université Montpellier, 34298 Montpellier, France. [4]Institut Régional du Cancer Montpellier, Université Montpellier, 34298 Montpellier, France. [5]Present address: Institute of Human Genetics, CNRS UMR9002 University of Montpellier, 34396 Montpellier, France. [6]These authors contributed equally: Karl Laviolette, Margot Zahm. ✉e-mail: olivier.joffre@inserm.fr

helper (Th) cell subsets. Thus, the molecular events that control signal-specific gene expression in T cells are essential for generating protective immune responses.

Upon activation, naive CD4[+] T cells first transcribe lineage-related but also non-related genes, before restricting gene expression to only the repertoire of molecular effectors specific to their differentiated phenotype (e.g. Th1, Th2, Th17)[1]. Thus, the establishment of a transcriptional signature in response to environmental cues is a progressive process driven by coordinated and highly specific gene-regulatory events. In CD4[+] T cells, transcriptional specificity is largely controlled by the combinatorial activities of STAT-family transcription factors, which are mobilized by lineage-specifying cytokines. It also involves chromatin-based regulations, which determine the repertoire of genomic elements amenable to activation and suppress the *cis*-regulatory regions associated with alternative fates[2,3]. Over the past decade, we and others exposed heterochromatin-dependent gene silencing in this process[4–6]. We showed that trimethylation of histone H3 lysine 9 (H3K9) by the lysine methyltransferase SETDB1 ensures Th2 cell lineage integrity by repressing the Th1 gene network[6]. Thus, from the plasma membrane to the nucleus, the molecular events that control gene expression in differentiating Th cells have been largely characterized.

It is not only activating signals that regulate T cell activation. To prevent autoimmune disorders and immunopathologies, T cells also integrate inhibitory signals that interfere with the molecular events that control gene expression in response to T cell receptor (TCR) engagement or lineage-specifying mediators. The immunosuppressive networks that control Th cell reactivity and functions are largely shaped by regulatory T cells (Treg), a subpopulation of CD4[+] T lymphocytes endowed with immunosuppressive functions and expressing the transcription factor forkhead box protein 3 (Foxp3)[7]. Their importance in immune system homeostasis is best exemplified by the severe systemic autoimmunity and lymphoproliferative disorders observed in Treg-deficient Scurfy mice and in human IPEX patients carrying non-functional or hypomorphic alleles of the *Foxp3* gene[8–11]. In addition, quantitative or functional defects of Treg are observed in several pathological immune responses, including Th cell-mediated autoimmunity and allergy[7]. Treg can directly interfere with T cell programming by producing immunosuppressive cytokines[7,12]. These mediators inhibit effector T cell functions and can even contribute to infectious tolerance, a process whereby Treg convey immunosuppressive properties to conventional T cells[13]. Extracellular adenosine generated by the ectonucleotidases CD39 and CD73, transfer of cAMP to effector T cells through gap junctions, IL-2 consumption, or killing of target cells through perforin- and granzyme-dependent mechanisms represent other Treg-mediated suppressive mechanisms. Treg can also act indirectly on T cells, by modulating dendritic cell maturation and function using co-inhibitory receptors such as CTLA-4 or Tim3. Thus, the multiple mechanisms used by Treg to inhibit naive CD4[+] T cell priming or effector and memory Th cell functions have been extensively identified[7,12]. However, how conventional T cells integrate and interpret these signals at the level of chromatin still remains largely unknown.

We and others recently reported that transcriptional specificity is largely controlled by heterochromatin-dependent gene silencing in differentiating Th cells[4–6,14]. We therefore hypothesized that Treg-mediated T cell suppression could also mobilize H3K9me3-HP1 epigenetic pathways. HP1 is a non-histone chromosomal protein that was initially described as a component of heterochromatin acting as a dosage-dependent modifier of position effect variegation[15,16]. HP1 is conserved through evolution, with homologs found from fission yeast to mammals[17]. The three different isoforms, called HP1α, HP1β and HP1γ in mammals, contain two conserved structural and functional motifs. The chromodomain binds to H3K9me3 and to the histone-fold of histone H3[18,19], whereas the chromoshadow domain allows HP1

interaction with the globular region of H3[20] and behaves as a homo- or hetero-dimerization module. In addition to their critical role in the folding and function of constitutive heterochromatin, the various HP1 isoforms are associated with different functions depending on their chromatin localization, the nuclear context, their post-translational modifications, or their binding partners[21]. Interestingly, despite their homology, HP1 variants exert at least in part non-redundant functions in T cells. With their binding partners, which include the corepressor TRIM28 and the lysine methyltransferases SETDB1 and SUV39H1, they are critically involved in CD4[+] and CD8[+] T cell programming in response to activating and polarizing signals[4–6,22].

In this work, we test whether HP1 variants and their interacting partners also regulate T-cell responses to immunosuppressive signals. Using a bone marrow allograft model, in which adoptively-transferred naive CD4[+] T cells coordinate graft rejection that Treg can inhibit[23,24], we first demonstrate that the three HP1 variants have no significant impact on the Th cell-mediated allogeneic response that leads to graft rejection. Interestingly, while we do not identify a role for HP1β in the crosstalk between Th cell and Treg, we observe that the α and γ isoforms critically and differently regulate Th cell behavior in response to suppressive signals. Unlike their wild-type counterparts, HP1α-deficient cells fail to efficiently repress their effector functions when exposed to Treg. Conversely, HP1γ-deficient cells are more sensitive to Treg-mediated suppression than wild-type cells. Mechanistically, our transcriptomic and epigenomic data indicate that the two HP1 variants control distinct sets of genes. HP1α is necessary for Treg-mediated silencing of the Th1 and Th17 gene networks. In contrast, HP1γ represses the TCR-induced expression of PD-1, LAG-3 and other immune checkpoints in differentiating Th cells. Thus, HP1γ-deficient Th cells express higher levels of these receptors and become more sensitive to inhibition by Treg. In conclusion, our findings reveal opposite roles for the heterochromatin regulators HP1α and HP1γ in Th cell sensitivity to immunosuppressive signals. These two non-histone chromosomal proteins may therefore be targeted in order to manipulate the dialogue between Treg and Th cells, depending on the pathophysiological context.

## Results
### Treg poorly control allogeneic responses mediated by HP1α-deficient T cells
To identify the molecular events that control the crosstalk between conventional (Tconv) and regulatory CD4[+] T cells, we used our previously described mouse model for bone marrow transplantation in which lethally irradiated B6 (H-2[b]) hosts are reconstituted with a 1:1 mixture of syngeneic (B6) and semi-allogeneic B6D2F1 (H-2[bd]) bone marrow (Fig. 1a)[23]. In this system, and in the absence of immune reconstitution, similar percentages of syngeneic and allogeneic cells are found in the blood three weeks after engraftment (Fig. 1b, c). In contrast, when naive Tconv are co-injected with bone marrow, the semi-allogeneic cells are fully eliminated and the retained syngeneic bone marrow allows the survival of the host (Fig. 1b, c). Importantly, the adoptive transfer of ex vivo expanded polyclonal Treg protects the transplant from rejection in a dose-dependent manner (Fig. 1b, c).

To analyze if heterochromatin-dependent gene silencing could regulate the sensitivity of Tconv to Treg-mediated suppression, we first used mice deficient for the non-histone chromosomal protein HP1α as a source of naive Tconv. We generated mice homozygous for a LoxP-flanked *Hp1a* allele and crossed them with *CMV*-cre transgenic mice. This strategy resulted in the complete absence of HP1α from naive CD4[+] T cells, with no compensation by overexpression of the two other HP1 isoforms, the methyltransferases targeting H3K9 or the transcriptional co-repressor TRIM28 (Figure. S1a–c). In the thymus of HP1α-deficient mice and control littermates, the relative proportion of the four main populations of thymocytes as well as the Tconv/Treg ratio were similar (Figure. S1d–f). We also failed to detect significant

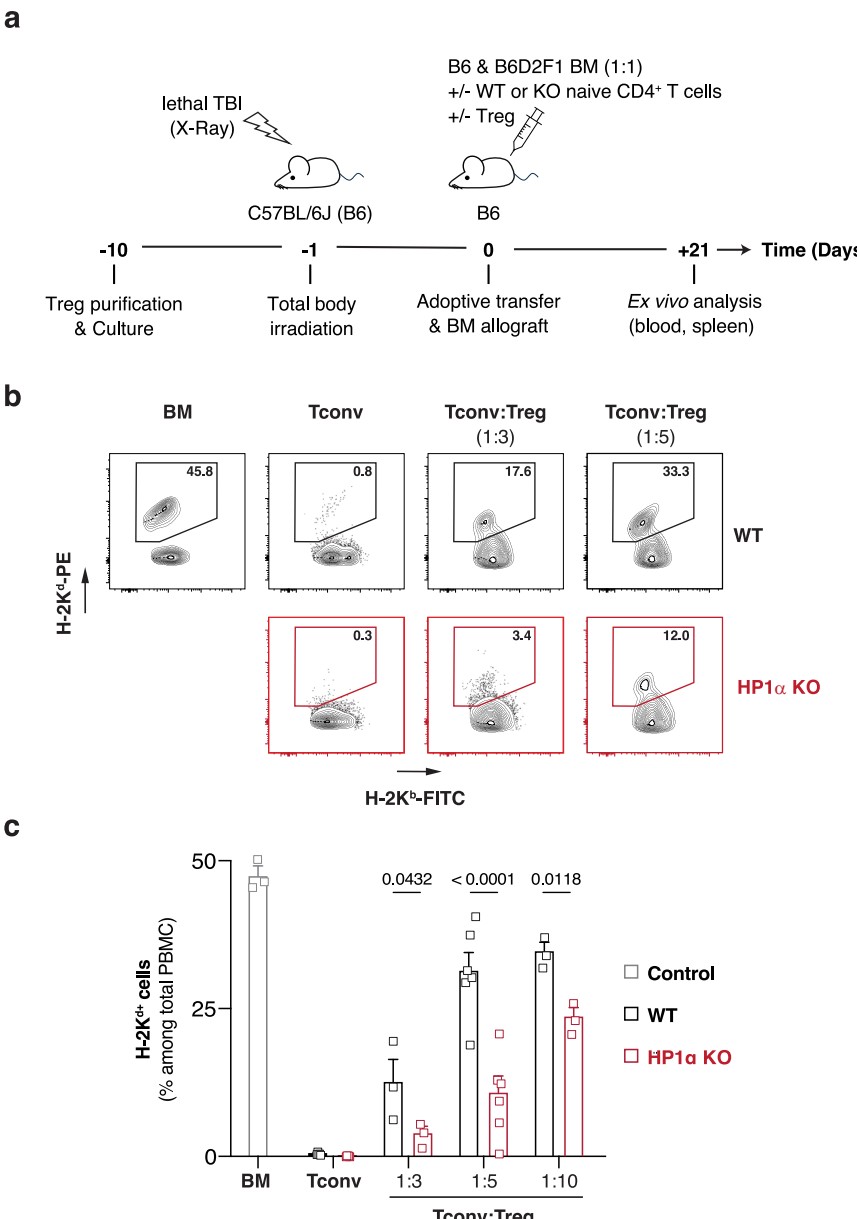

**Fig. 1 | Treg are less effective in regulating allogeneic responses mediated by HP1α-deficient T cells than by their wild-type counterparts.** Lethally-irradiated B6 mice were grafted with a 1:1 mixture of B6 (H-2$^b$) and B6D2F1 (H-2$^{bd}$) bone marrow and co-injected with WT or HP1α-deficient naive CD4$^+$ T cells alone or in the presence of ex vivo expanded WT Treg. To control engraftment, a group of mice was also injected solely with the bone marrow cell mixture. **a** Experimental model. **b** Representative dot-plots showing the frequency of syngeneic (H-2K$^{b+}$) and semi- allogeneic (H-2K$^{bd+}$) cells in the blood 21 days after engraftment. **c** Percentage of semi-allogeneic (H-2K$^{d+}$) cells among peripheral blood mononuclear cells (PBMC) 21 days after engraftment. Data show mean ± SEM of 3 to 6 independent experiments. Each symbol represents the mean of an experiment. P values were calculated using multiple unpaired t-tests with Holm-Šidák correction. Source data are provided in the Source data file.

differences between peripheral CD4$^+$ and CD8$^+$ T cells from the two genotypes (Figure. S1g–p), indicating that HP1α deficiency did not affect the global homeostasis of the αβ T cell compartment. We then isolated naive Tconv from HP1α-deficient mice (Figure. S1p, q) and co-injected them with the mixture of syngeneic and semi-allogeneic bone marrow. Three weeks later, as detected with wild-type Tconv, we observed complete rejection of the semi-allogeneic transplant (Fig. 1b, c). This result shows that HP1α-deficiency does not impair naive CD4$^+$ T cell priming and Th cell functions. To test if HP1α regulates Treg-mediated suppression of Tconv function, we then co-injected wild-type or *Hp1a$^{-/-}$* naive CD4$^+$ T cells together with increasing numbers of ex vivo expanded polyclonal Treg (Figure. S1r). Irrespective of the number of Treg transferred, the frequency of semi-

allogeneic cells detected in the blood was significantly higher in mice in which graft rejection was mediated by wild-type Tconv than in animals injected with HP1α-deficient cells (Fig. 1b, c). These results show that HP1α-deficient Tconv are less sensitive to Treg-mediated suppression than wild-type cells.

## HP1α-deficient Th1 and Th17 cells are resistant to Treg-mediated suppression

To identify the cellular functions regulated by HP1α, we then analyzed T cell programming ex vivo by flow cytometry. Adoptive transfer of T lymphocytes into lymphopenic mice leads to lymphopenia-induced proliferation[25,26]. To focus on donor-specific Tconv, we analyzed Vβ6$^+$ T cells, which recognize the MMTV-7 encoded superantigen expressed

by donor (but not by host) cells[27]. As expected, in mice adoptively transferred with wild-type Tconv alone, donor-specific T cells produce large amount of the Th1 cytokine IFN-γ (Fig. 2a, b). We also observed that the allogeneic T-cell response was characterized by the differentiation of a small proportion of IL-17A-producing cells (Fig. 2a, c). In the absence of Treg, HP1α-deficient cells phenocopy their wild-type counterparts and acquire a Th1- or Th17-like phenotype (Fig. 2a-d). This observation is consistent with the similar ability of wild-type and mutant cells to induce graft rejection and to differentiate into Th1 or Th17 effector cells in vitro. Control and HP1α-deficient cells indeed showed similar activation and proliferation upon T cell receptor triggering (Figure. S2a-e), and comparable Th1 and Th17 priming when exposed to lineage-specifying cytokines (Figure. S2f-o). Altogether, these data support the conclusion that HP1α does not control Tconv activation and differentiation.

In contrast, exposure of wild-type vs. Hp1a[-/-] Tconv to Treg-mediated suppressive signals in vivo revealed marked differences. While the production of inflammatory cytokines by wild-type Tconv is effectively inhibited by Treg, HP1α-deficient cells are at least in part refractory to their suppressive environment and continue to produce high levels of IFN-γ and IL-17A (Fig. 2a-d). These results suggest that HP1α relays immunosuppressive signals at the chromatin level to repress the Th1 and Th17 effector programs.

However, HP1α-deficiency in Tconv has no detectable impact on Treg programming (Fig. 2e) and does not affect all Treg functions. The frequency and absolute number of (TCR Vβ6[+]) Tconv among splenocytes were similar in mice injected with wild-type vs. Hp1a[-/-] Tconv (Fig. 2f-h), showing that Treg-induced contraction of the splenic T cell compartment was not affected. HP1α-deficiency also did not alter the Treg-induced conversion of Tconv into anergic and regulatory T cells (Fig. 2i, j). Therefore, whereas HP1α is required for efficient Treg-mediated suppression of Th1 and Th17-differentiation, it does not appear to be involved in the suppression of T cell accumulation or in Tconv differentiation into anergic or regulatory cells.

## HP1α directly modulates the chromatin of Th1 and Th17 genes

To characterize the gene networks that HP1α regulates, we next used deep total RNA-sequencing (RNA-seq) to analyze the transcriptomes of wild-type and mutant cells. Despite deleting a key component of heterochromatin, the transcriptomes of naive CD4[+] T cells freshly isolated from wild-type and Hp1a[-/-] mice showed no significant difference (Fig. 3a). In contrast, the expression of 176 genes was significantly upregulated in HP1α-deficient Tconv isolated from mice adoptively transferred with CD4[+] T cells (and the mix of bone marrow) only. To determine whether this cluster was associated with specific biological functions, we next performed gene ontology (GO)-based over-representation analysis (ORA). The results indicate that HP1α-deficient Tconv overexpressed genes that regulate T cell activation (Figure. S3a). Gene set enrichment analysis (GSEA) confirmed these results and further associated this gene set with Th17 cell-dependent immunity (Figure. S3b). So, even if we did not detect functional differences between control and mutant cells in vitro (Figure. S2) and ex vivo using flow cytometry-based strategies (Fig. 2), our transcriptomic data reveals that mutant Tconv from the BM-chimeras have an exacerbated activated phenotype compared to their wild-type counterparts. However, the greatest differences in programming were revealed when Tconv were exposed to a Treg-mediated suppressive environment in vivo. In this context, HP1α-deficient cells over-expressed 348 genes and under-expressed 365 genes. As expected from our ex vivo flow cytometry analyses (Fig. 2), HP1α-deficiency increased the expression of gene clusters functionally involved in various Th cell functions (Fig. 3b, c). It also correlated with an impaired mobilization of the regulatory circuits, which inhibit T cell-programming (Fig. 3c) and of gene sets involved in exhaustion (Fig. 3d).

HP1α can regulate heterochromatin-dependent gene silencing. Therefore, we then used "assay for transposase accessible chromatin using sequencing" (ATAC-Seq) to determine to what extent the alteration in gene expression in HP1α-deficient vs. wild-type cells was linked to modifications of gene accessibility. Interestingly, we only detected differentially accessible regions when we compared wild-type and mutant Tconv exposed to Treg-dependent suppressive pathways (Fig. 3e). When we intersected the coordinates of the genes flanking genomic regions more accessible in HP1α-deficient Tconv with those from the genes they overexpressed, we identified 54 genes (Supplementary Data 1) that were at the same time more accessible and more expressed in mutant cells (Fig. 3f). In a wild-type context, these genes were upregulated in Tconv upon activation, and strongly inhibited when these cells were exposed to Treg (Fig. 3g). In the absence of HP1α, their dynamics of expression were similar except that they failed to be efficiently repressed in response to immunosuppressive signals. As expected, ORA and GSEA analyses revealed that Th1 and Th17 signature genes were overrepresented within this gene set (Fig. 3h-j and S3d). Altogether, these results show that Treg-dependent repressive signals mobilize HP1α to repress a cluster of genes critically involved in Th cell functions. Of note, analysis of the genes that were less accessible and less expressed in mutant as compared to wild-type cells did not reveal any relevant biological pathways (Figure. S3e-g). Accessibility and expression of these genes are most probably indirectly regulated by HP1α.

HP1α-deficiency therefore enables donor-specific murine Tconv to maintain their effector functions in a Treg-mediated immunosuppressive environment. To assess whether targeting this molecule could be of clinical interest, we next compared the transcriptomic signature induced by HP1α-deletion in murine CD4[+] T cells to the transcriptome of tumor-infiltrating T lymphocytes (TILs) isolated from patients with advanced melanoma[28]. Single-cell profiling of intratumoral human CD4[+] T cells first allowed us to identify four functionally distinct Tconv subsets (Fig. 3k). Interestingly, these four T cell populations are differentially represented in patients according to their ability to respond to immune checkpoint inhibitors (i.e. anti-PD1 and/or anti-CTLA-4 antibodies) (Fig. 3l). While effector T cells (Teff) accumulate in responder individuals (R) at the expense of exhausted cells (Tex), the Teff/Tex ratio drops in non-responder patients (NR, Fig. 3l, m). Despite immune checkpoint blockade, tumor-specific CD4[+] T cells from NR patients therefore failed to maintain their effector functions and acquired an exhausted phenotype. To test whether HP1α inhibition could potentially block or reverse this process, we finally compared the transcriptomic signature induced by HP1α-deficiency in allospecific murine CD4[+] T cells to those of intratumoral human Teff and Tex. Interestingly, we observed a significant enrichment of genes downregulated in HP1α-deficient cells among those that define Tex identity (Fig. 3n). These data suggest that inhibiting HP1α in tumor-specific human CD4[+] T cells could, as it does in allospecific murine Tconv (Fig. 3c, d), repress the establishment of the exhaustion program and thus promote a protective immune response.

## Treg better control allogeneic responses when the SUV39H1-HP1γ axis is inactivated

We next used our mouse model for bone marrow transplantation to test if other regulators of heterochromatin are mobilized by Treg-dependent suppressive signals. Given the sequence and functional homology between the α and β isoforms, we were quite surprised to find that HP1β-deficient Tconv were as efficiently suppressed by Treg as wild-type cells (Figure. S4a, b). We then tested the involvement of HP1γ in the susceptibility of allospecific Tconv to Treg-mediated suppression. We generated mice carrying a conditional ready Hp1g (Hp1g[flox]) allele and crossed them with Cd4-cre mutant mice to generate a conditional HP1γ-deficient (Hp1g[-/-]) strain. Naive CD4[+] T cells isolated from these mice exhibited almost complete deletion of HP1γ, with no

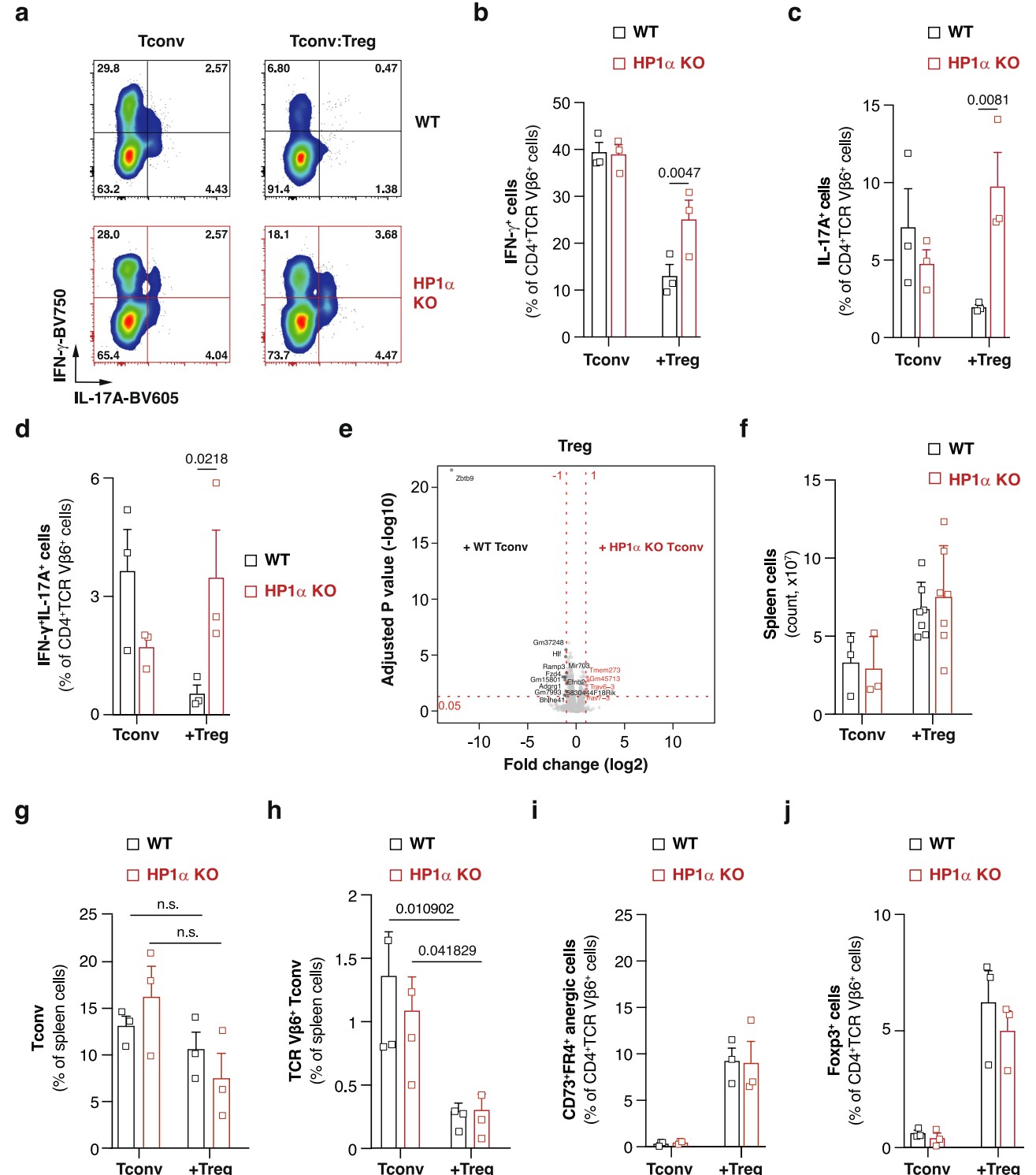

**Fig. 2 | HP1α-deficient Th1 and Th17 cells are poorly repressed by Treg.** Lethally-irradiated B6 mice were grafted with a 1:1 mixture of B6 and B6D2F1 bone marrow and co-injected with WT or HP1α-deficient naive CD4+ T cells alone or in the presence of ex vivo expanded WT Treg. **a** Representative dot-plots showing 21 days after engraftment the percentage of WT and HP1α KO TCR Vβ6+CD4+ T cells producing IFN-γ and IL-17A. **b–d** Percentage of WT or HP1α KO TCR Vβ6+CD4+ T cells producing IFN-γ (b), IL-17A (c) or both (d) 21 days after engraftment. **e** Volcano plot showing results of differential gene expression analyses between Treg that had been co-injected with WT or HP1α KO naive CD4+ T cells. Red and black dots represent genes with higher expression in Treg co-injected with HP1α KO or WT cells, respectively. Gray dots represent genes that failed to reach the FDR threshold of 0.05 and the absolute log2 fold change threshold of 1. **f** Absolute number of spleen cells 21 days after engraftment. Data are mean ± SD of three (Tconv) or seven (Tconv + Treg) biological replicates from two independent experiments. **g** Percentage of Tconv, defined as CD4+CD45.1‾CD45.2+H-2K^d‾Thy1.1‾, in the spleen 21 days after engraftment. Data are mean ± SEM of three independent experiments. **h** Percentage of allospecific Tconv, defined as CD4+CD45.1‾CD45.2+H-2K^d‾Thy1.1‾TCRVβ6+, in the spleen 21 days after engraftment. Data are mean ± SEM of three independent experiments. **i, j** Percentage of CD73+FR4+ (i) or Foxp3+ (j) cells among WT or HP1α KO TCR Vβ6+CD4+ T cells, as determined in the spleen 21 days after engraftment. **b–d** and **g–j** Data show mean ± SEM of 3 independent experiments. Each symbol represents the mean of an experiment. *P* values were calculated using multiple unpaired t test with Holm-Šidák correction. Source data are provided in the Source data file.

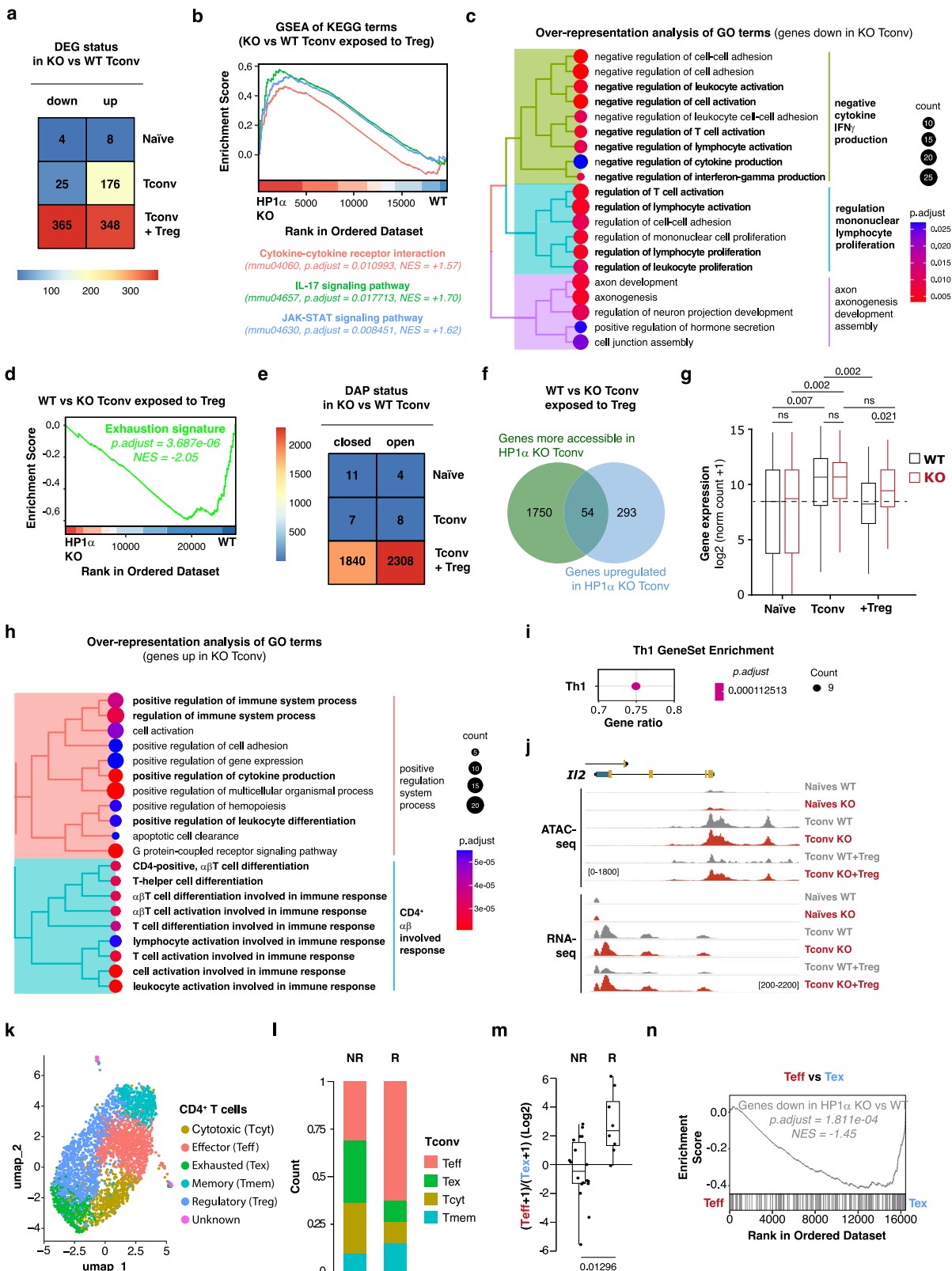

compensation by overexpression of the two other HP1 isoforms, the methyltransferases targeting H3K9 or the transcriptional co-repressor TRIM28 (Figure. S4c-e). It was proposed that the SUMO E3 ligase PIAS1 negatively regulates Treg development by maintaining a repressive chromatin state at the Foxp3 promoter through the recruitment of HP1γ[29]. We therefore analyzed if deleting HP1γ affected the Treg compartment and did not detect any significant difference in T cell-

development in the thymus. We also did not detect any impact of the deletion on peripheral αβ T cell homeostasis and programming (Figure. S4f-r). In our bone marrow-transplantation model, highly-purified HP1γ-deficient Tconv (Figure. S4s) efficiently rejected the semi-allogeneic graft, suggesting that HP1γ deletion did not affect the allogeneic T cell response (Fig. 4a, b). Unexpectedly, Treg protected better from allograft rejection mediated by HP1γ-deficient than by wild-type

**Fig. 3 | HP1α translates Treg-dependent suppressive signals at the chromatin level. a** Differentially-expressed genes (DEG) between WT and HP1α-deficient T cells. **b** GSEA of KEGG pathways performed using transcriptomes of HP1α-deficient and WT Tconv exposed to Treg. **c** GO enrichment analyses of genes more highly expressed in WT Tconv than in HP1α KO Tconv exposed to Treg. **d** GSEA of exhaustion signature genes performed using transcriptomes of HP1α-deficient and WT Tconv exposed to Treg. **e** Number of differentially-accessible peaks (DAP) between WT and HP1α-deficient T cells. **f** Relationship between the genes more highly expressed and the genes associated with peaks more open in HP1α KO Tconv than in WT Tconv exposed to Treg. **g** Expression levels of the 54 genes selected from the intersection in **f**. Horizontal bars represent the median, and the top and bottom of the boxes the upper and lower quartiles, respectively. The whiskers go from the minimum to the lower quartile and from the upper quartile to the maximum. Statistical significance was calculated using the Pairwise Wilcoxon

Rank Sum Test (two-tailed). **h** GO enrichment analyses of the 54 genes selected from the intersection in **f**. **i** Th1 signature enrichment analyses of the 54 genes selected from the intersection in **f**. **j** ATAC–seq and RNA-seq tracks at the *Il2* locus. **k** UMAP plot of tumor-infiltrating CD4[+] T lymphocytes from melanoma patients[28]. Cells are colored based on 6 clusters defined using the SNN algorithm. **l** Relative proportions of CD4[+] T cell subsets in patients who responded (R) or not (NR) to immunotherapy. **m** Teff:Tex ratio in R and NR patients. Statistical significance was calculated using Wilcoxon rank sum Test (two-tailed). Horizontal bars represent the median, and the top and bottom of the boxes the upper and lower quartiles, respectively. The whiskers go from the minimum to the lower quartile and from the upper quartile to the maximum. **n** GSEA of genes downregulated in HP1α-deficient *vs* control Tconv performed using transcriptomes of human Teff and Tex. Significance was estimated using rank-based gene permutations. Source data are provided in the Source data file.

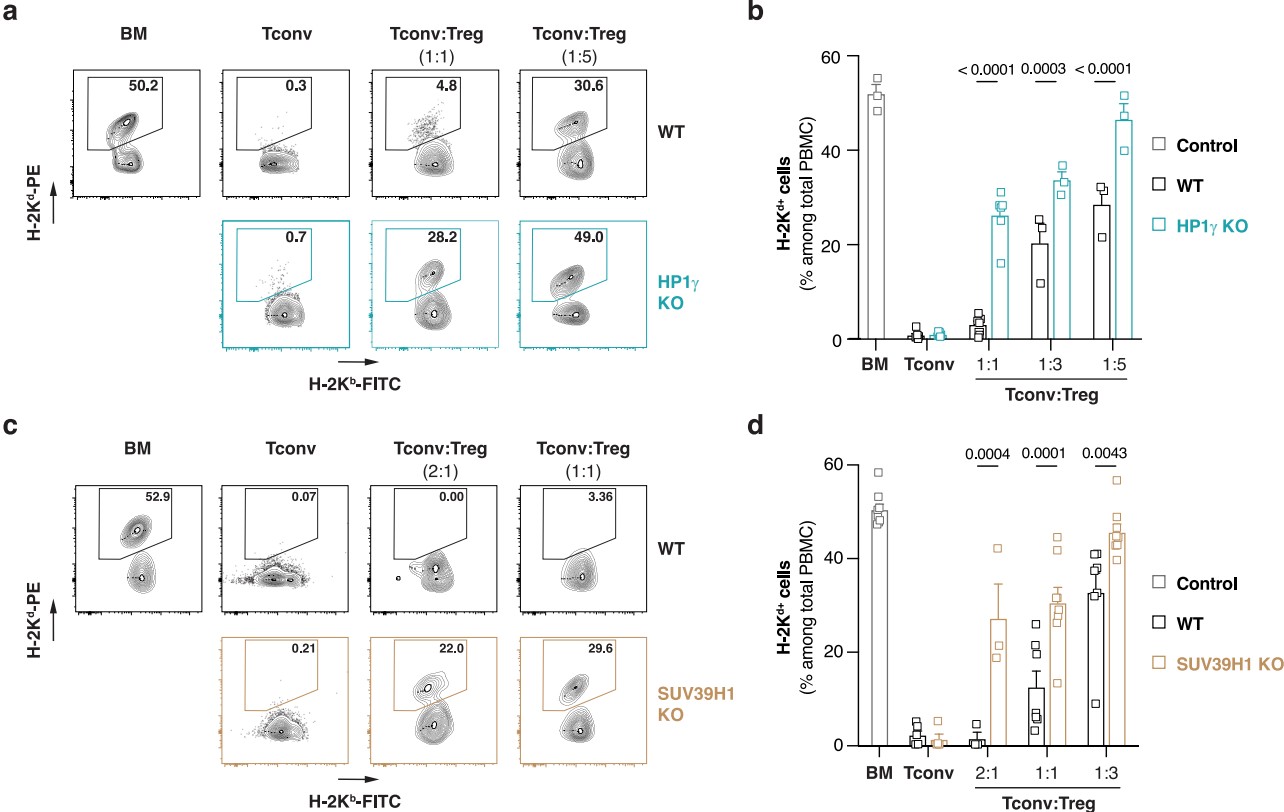

**Fig. 4 | The SUV39H1-HP1γ axis controls alloreactive T cell susceptibility to Treg-mediated suppression.** Lethally-irradiated B6 mice were grafted with a 1:1 mixture of B6 and B6D2F1 bone marrow and co-injected with WT or HP1α KO (**a, b**) or SUV39H1 KO (**c, d**) naive CD4[+] T cells alone or in the presence of ex vivo expanded WT Treg. To control engraftment, a group of mice was also injected solely with the bone marrow cell mixture. **a, c** Representative dot-plots showing the

frequency of syngeneic (H-2K[b+]) and semi-allogeneic (H-2K[bd+]) cells in the blood 21 days after engraftment. **b, d** Percentage of semi-allogeneic (H-2K[d+]) cells among PBMC 21 days after engraftment. Data show mean ± SEM of 3 to 8 independent experiments. Each symbol represents the mean of an experiment. *P* values were calculated using multiple unpaired t test with Holm-Šidák correction. Source data are provided in the Source data file.

Tconv. We obtained similar results with Tconv deficient for the lysine methyltransferase SUV39H1, which catalyzes H3K9me3 deposition on the chromatin (Fig. 4c, d). Altogether, these results show that the SUV39H1-HP1γ epigenetic pathway interferes with Treg-mediated inhibition of Tconv in vivo.

## HP1γ-deficiency favors Th cell repression and conversion into anergic and regulatory T cells

To understand how HP1γ-deficiency in Tconv leads to more efficient Treg-mediated repression of allogeneic T cell responses, we first tested if this chromatin-organizer regulates Th cell priming. We isolated wild-type and *Hp1g⁻/⁻* naive CD4[+] T cells and compared their activation and differentiation following T cell receptor engagement and exposure to

Th1 or Th17 lineage-specifying factors. After two days of culture, the upregulation of the activation markers CD25 and CD69 was similar between control and mutant cells (Figure. S5a, b). When cultured under Th1- or Th17-polarizing conditions, wild-type and HP1γ-deficient T cells also expressed similar levels of the lineage-determining master regulators T-bet and RORγt and similarly produced the Th1 and Th17 signature cytokines IFN-γ and IL-17/GM-CSF (Figure. S5c-i). These observations were confirmed in vivo. In our bone marrow transplantation model, when transferred alone, wild-type and HP1γ-mutant Tconv mounted similar T cell-responses dominated by a Th1-phenotype (Fig. 5a-c). Consistent with the more efficient Treg-mediated protection of the allograft from rejection by HP1γ-deficient T cells than by wild-type T cells, we observed that Treg more effectively

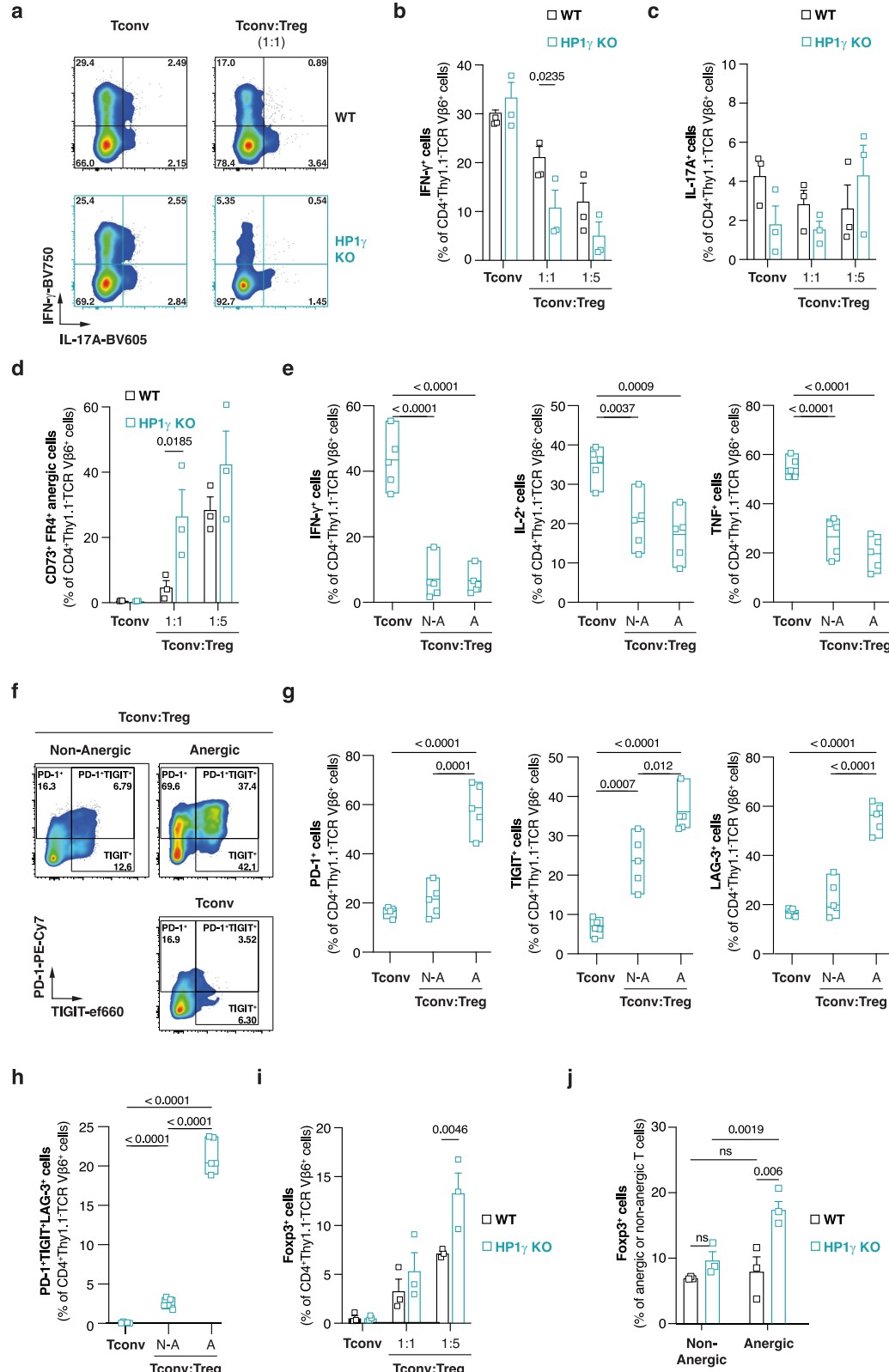

inhibited IFN-γ production by mutant Tconv than their wild-type counterparts (Fig. 5a, b). We confirmed the higher sensitivity of *Hp1g⁻/⁻* Tconv to Treg-mediated inhibition using a classical in vitro Treg suppression assay (Figure. S5j). In vivo, the better repression of *Hp1g⁻/⁻* Tconv correlated with enhanced accumulation of donor-reactive anergic mutant cells (Fig. 5d). In addition to expressing CD73 and FR4 markers, these anergic cells appeared functionally exhausted.

They fail to produce IFN-γ, IL-2 and TNF (Fig. 5e) and a large fraction of them co-express the immune checkpoints PD-1, LAG-3 and TIGIT (Fig. 5f-h). At higher doses of Treg, we also observed that HP1γ-deficiency correlated with higher differentiation of donor-specific Foxp3⁺ regulatory T cells (Fig. 5i). We found more HP1γ-deficient Foxp3⁺ cells among CD73⁺FR4⁺ anergic than among non-anergic T cells (Fig. 5j), suggesting that preferentially anergic cells may differentiate

**Fig. 5 | Th cell inhibition and conversion into anergic or regulatory T cells is more efficient in the absence of HP1γ.** Lethally-irradiated B6 mice were grafted with a 1:1 mixture of B6 and B6D2F1 bone marrow and co-injected with WT (**a-d** and I, j) or HP1γ-deficient (**a–j**) naive CD4[+] T cells alone or in the presence of ex vivo expanded WT Treg. **a** Representative dot-plots showing 21 days after engraftment the percentage of WT and HP1γ KO TCR Vβ6[+] Tconv producing IFN-γ and IL-17A. **b, c** Percentage of WT or HP1γ KO TCR Vβ6[+] Tconv producing IFN-γ (**b**) or IL-17A (**c**) 21 days after engraftment. **d** Percentage of CD73[+]FR4[+] cells among WT or HP1γ KO TCR Vβ6[+] Tconv, as determined in the spleen 21 days after engraftment. **e** Percentage of cytokine-producing cells among HP1γ KO TCR Vβ6[+] Tconv 21 days post-engraftment. The analysis was performed on total Tconv from mice injected with Tconv only, or on anergic (A) or non-anergic (N-A) Tconv from mice injected with Tconv and Treg. (**f**) Representative dot-plots showing 21 days after

engraftment the percentage of HP1γ KO TCR Vβ6[+] Tconv expressing PD-1 and TIGIT. **g, h** Percentage of PD-1[+], TIGIT[+], LAG-3[+] (**g**) or PD-1[+]TIGIT[+]LAG-3[+] (**h**) cells among HP1γ KO TCR Vβ6[+] Tconv 21 days after engraftment. **i** Percentage of Foxp3[+] cells among WT or HP1γ KO TCR Vβ6[+] Tconv, as determined in the spleen 21 days after engraftment. **j** Percentage of spleen Foxp3[+] cells among anergic or non-anergic WT and HP1γ KO TCR Vβ6[+] Tconv 21 days after engraftment. **b–d** and **I, j** Data are represented as mean ± SEM of three independent experiments. Each symbol represents the mean of an experiment. *P* values were calculated using multiple unpaired t test with Holm-Šidák correction. **e, g, h** Floating bar charts represent the mean of the biological replicates as well as the minimum and maximum values. Each symbol represents individual biological replicates. *P* values were calculated using unpaired t test (two-tailed). Source data are provided in the Source data file.

into Treg, in agreement with a recent report[30]. Altogether, these results indicate that HP1γ-deficiency favors the acquisition of a state of functional unresponsiveness when Tconv are exposed to Treg-mediated immunosuppressive signals.

## HP1γ is a negative regulator of a network of genes functionally associated with T-cell exhaustion

To delineate gene networks that the SUV39H1-HP1γ epigenetic pathway regulates, we then applied RNA-seq to analyze the transcriptomes of wild-type and HP1γ-deficient T cells. Since our functional analyses revealed differences between wild-type and mutant cells only when Tconv were exposed to immunosuppressive signals, we focused our attention on the genes that were differentially expressed between *Hp1g*[-/-] and *Hp1g*[+/+] Tconv isolated from mice that had been co-injected with Treg. The weighted correlation network analysis of the genes overexpressed in HP1γ-deficient Tconv compared with wild-type Tconv in the presence of Treg identified four clusters with different expression patterns (Fig. 6a). Among them, cluster 2 was of high interest because it contained almost 75% of the genes differentially expressed between wild-type and HP1γ-deficient Tconv exposed to Treg. When we analyzed their dynamics in wild-type cells, we observed that they were significantly induced by activation and that exposure to Treg did not affect their expression. (Fig. 6b). Given that at the Tconv/Treg ratio used in this experiment, the allogeneic response mediated by wild-type cells was not effectively repressed, these observations suggested that cluster 2 genes may be involved in T cell activation and function. However, these genes were also strongly induced in differentiating HP1γ-deficient Tconv, and were even more highly expressed when these cells were converted into anergic or regulatory T cells by Treg (Fig. 6b). To identify the biological function of cluster 2 genes, we performed ORA and GSEA analyses. The results revealed a strong enrichment in biological processes associated with T cell suppression and exhaustion (Fig. 6c, d). To test if HP1γ regulates these genes through heterochromatin-dependent gene silencing, we next determined their accessibility by ATAC-seq in control and mutant T cells. Our analysis revealed that 151 cluster 2 genes (Supplementary Data 2) were at the same time more accessible and more expressed in Treg-exposed HP1γ-deficient Tconv than in wild-type cells (Fig. 6e, f). Among them, we again identified a cluster critically involved in T cell exhaustion and anergy. It included the genes encoding the immunosuppressive cytokine IL-10, the immune checkpoints TIM3, LAG3 and PD-1, and the transcription factor TOX (which has been proposed to transcriptionally and epigenetically programs CD8[+] T cell exhaustion)[31]. Interestingly, even if these genes were only highly expressed when mutant Tconv were exposed to Treg (Fig. 6a, b), our ATAC-seq data show that they were already partially open in mutant Tconv in the absence of immunosuppression (Fig. 6g). This observation suggested that the HP1γ-dependent epigenetic mechanisms that negatively regulate this gene network are already at work in activated T cells. To test this hypothesis, we isolated wild-type and mutant naive CD4[+] T cells and stimulated them in vitro with anti-CD3 antibody. As

expected, the frequency of PD-1 and LAG-3-expressing cells and the average expression of these immune checkpoints were higher in HP1γ-deficient cells than in control cells (Fig. 6h, k). Since SUV39H1 KO cells phenocopy HP1γ-deficient cells in vivo, we finally tested whether similar results could be obtained by exposing wild-type cells to chaetocin, a SUV39H1 inhibitor. Remarkably, acute inhibition of this lysine methyltransferase increased PD-1 and LAG-3 expression in activated CD4[+] T cells in a manner similar to that observed in HP1γ-deficient cells (Fig. 6l-n). Therefore, in differentiating Th cells SUV39H1 and HP1γ repress a network of genes functionally involved in T cell exhaustion and anergy. Our functional data suggest that their deregulation in HP1γ-deficient cells could promote T cell repression in a suppressive environment, thereby contributing to the induction of a state of immune tolerance.

## HP1γ-deficiency promotes repression of human CD4[+] T cells

We next tested whether HP1γ-deficiency could also sensitize human CD4[+] T cells to immunosuppressive signals. To do so, we set up a mouse model of xenogeneic graft-versus-host-disease (xGvHD) in which sublethally-irradiated immunodeficient NOD-*scid* IL2Rgamma[null] (NSG) hosts are injected with human naive CD4[+] T cells alone or in the presence of human Treg (Fig. 7a). In this system, and in the absence of suppressive cells, a strong xenogeneic Th1 response develops in the weeks following injection. Xenoantigen-activated human CD4[+] T cells produce large amount of IFN-γ and GM-CSF (Fig. 7d-f), acquire cytotoxic capacity (Fig. 7g), and express T-bet, the master regulator of the Th1 lineage (Fig. 7h). This pathogenic immune response can be inhibited by co-injection of large numbers of Treg[32] but at a Treg:Tconv ratio of 2:1 the suppressor cells fail to efficiently repress xenospecific wild-type T cells (Fig. 7d-h). To test whether HP1γ-deficiency could restore Tconv-susceptibility to Treg-mediated suppression, we repeated the experiment but using human naive CD4[+] T cells in which HP1γ-expression had previously been inactivated by CRISPR-Cas9 (Fig. 7b, c). Remarkably, while HP1γ-deficiency had no significant impact on the xenogeneic response in the absence of suppressor cells (Fig. 7d-h), it did enable Treg to effectively repress the Th1 response. In contrast to wild-type cells, which were poorly sensitive to the immunosuppressive activity of Treg in this setting, mutant cells almost completely repressed IFN-γ secretion and Granzyme B production (Fig. 7e, g). Interestingly, some rare but detectable mutant cells also produce the anti-inflammatory cytokine IL-10 and express the transcription factor Foxp3 in the presence of suppressor cells (Fig. 7i, j). Taken together, these in vivo data suggest that HP1γ reduces the susceptibility of human CD4[+] T cells to immunosuppressive signals, and that targeting this molecule could be clinically relevant for repressing pathogenic T cell responses in bone marrow transplant individuals or in patients suffering from autoimmunity.

## Discussion

Since the discovery of Treg in the late 1990s, the immunoregulatory pathways mobilized by these cells to prevent immunopathology and regulate protective immune responses have been extensively

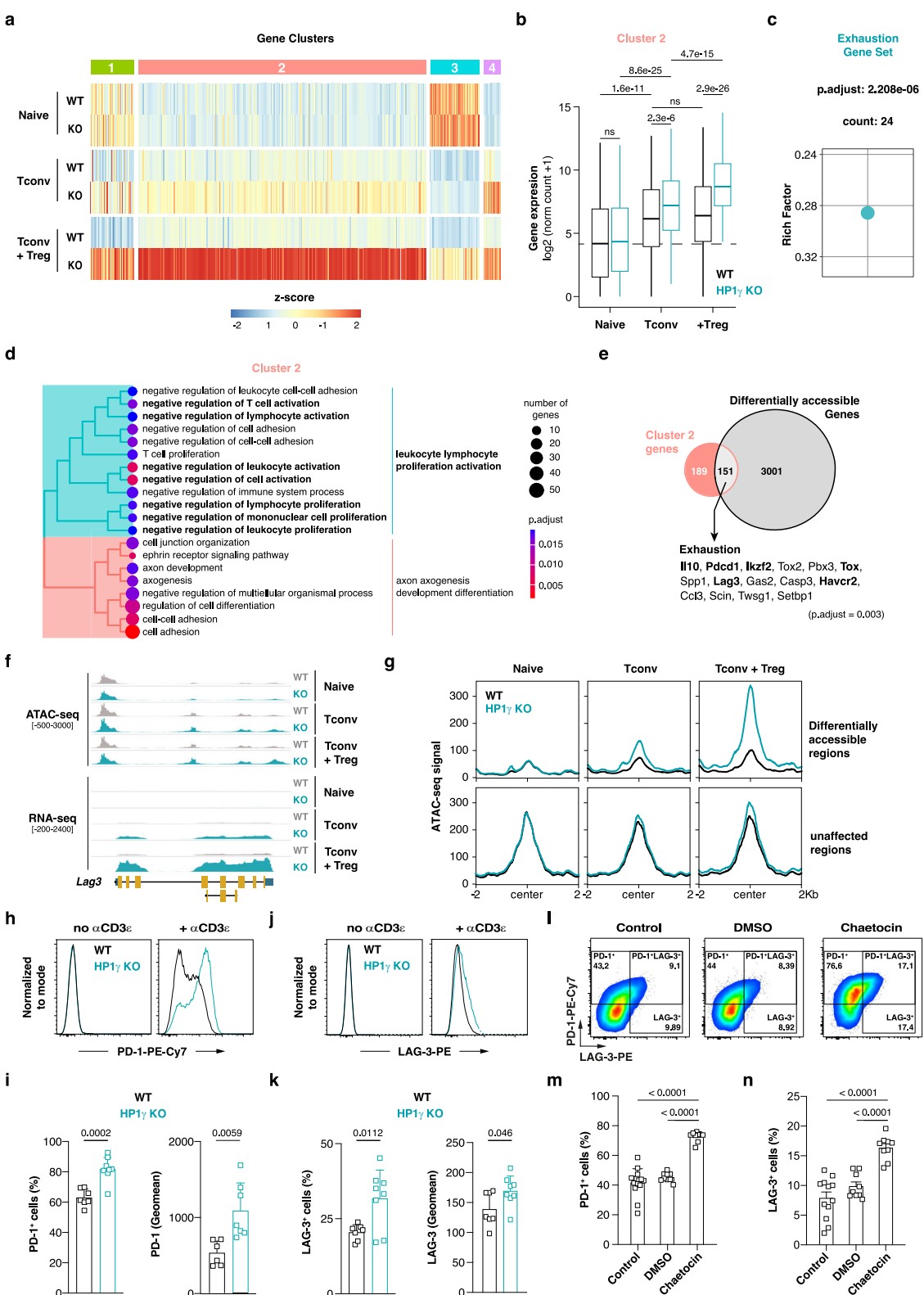

characterized. The fate of immune cells exposed to suppressive signals has also been largely documented. For T cells, Treg-mediated suppression leads, depending on the pathophysiological context and the repressive pathways involved, to anergy, exhaustion, apoptosis or differentiation into Treg by a process known as infectious tolerance[33,34]. In this work, we attempted to identify, downstream of the signaling networks mobilized by Treg-dependent signals, the chromatin molecules that reshape the epigenetic landscape of Tconv and thus coordinate their reprogramming. In line with our initial working hypothesis, we showed that HP1α is critically involved in Treg-mediated Th cell suppression through a process implicating heterochromatin-dependent silencing of Th1 and Th17 genes. More surprisingly, we also demonstrated that one of its paralogs, HP1γ, acts as a gatekeeper for a network of genes associated with T cell anergy and exhaustion. In response to TCR engagement, immune checkpoints such as PD1 and LAG3 are over-expressed in HP1γ-deficient Tconv, which potentiates their sensitivity to immunosuppressive signals and ultimately the control of the immune response by Treg. In conclusion,

**Fig. 6 | HP1γ negatively controls immune checkpoint expression in effector Th cells. a** Pattern of expression of genes overexpressed in Treg-exposed HP1γ KO Tconv compared with WT Tconv. (**b**) Cluster 2 genes expression in indicated Tconv populations. Horizontal bars represent the median, and the top and bottom of the box the upper and lower quartile, respectively. The whiskers go from the minimum to the lower quartile and from the upper quartile to the maximum. Statistical significance was calculated using the Pairwise Wilcoxon Rank Sum Test (two-tailed). **c** Exhaustion signature enrichment analyses of cluster 2 genes. **d** GO enrichment analyses of cluster 2 genes. **e** Euler diagram showing the relationship between cluster 2 genes and those associated with peaks more open in Treg-exposed HP1γ KO Tconv than WT Tconv. Exhaustion signature enrichment analysis was run on genes common to both groups. **f** ATAC-seq and RNA-seq tracks at the *Lag3* locus. **g** ATAC-seq signal at peaks associated with cluster 2 genes. Peaks were divided into two sets according to whether or not they were differentially open in WT and KO T cells exposed to Treg. **h**–**k** PD-1 and LAG3 expression on naive CD4[+] T cells before or after two days of culture with anti-CD3 antibody. **h, j** Representative histograms showing the expression of PD-1 (**h**) or LAG3 (**j**) in naive and activated T cells. **i, k** Percentage of activated T cells expressing PD-1 (**i**) or LAG3 (**k**) (left) and average immune checkpoint expression per cell (right). **l** Representative dot-plots showing PD-1 versus LAG3 expression by ex vivo stimulated CD4[+] T cells with or without (control) DMSO or chaetocin. **m, n** Percentage of CD4[+] T cells, either ex vivo activated and treated or not (control) with DMSO or chaetocin, expressing PD-1 (**m**) or LAG-3 (**n**). **l, k, m, n** Data show mean ± SD of biological replicates from 4 independent experiments. Each symbol represents an individual biological replicate. *P* values were calculated using unpaired t test (two-tailed). Source data are provided in the Source data file.

we demonstrated that heterochromatin-dependent epigenetic pathways critically regulate Th-cell susceptibility to Treg-mediated suppression, and we identified two epigenetic players whose expression may be manipulated to strengthen or suppress immune responses.

Our data indicate that HP1α and HP1γ regulate the dialogue between Tconv and Treg in opposite ways. Whereas HP1α coordinates the effect of immunosuppressive signals at the chromatin level, HP1γ negatively regulates the expression of immune checkpoints and thus limits T-cell suppression. However, these two paralogs have the same structure and functional domains. From the N-terminal (NTE) to the C-terminal (CTE) extension, they hold a chromodomain (CD), which specifically binds H3K9me2/3, a hinge region, which interacts with nucleic acids, and a chromoshadow domain (CSD), which allows homo- or heterodimerization with binding partners that contain a PXVXL motif. How then can we explain the opposite roles of the two molecules? While the CSD and CD domains of the different paralogs have a sequence homology of over 70%, they do not share exactly the same functional properties. It has, for example, been documented that domain-swapping between different paralogs can affect their respective nuclear localization and biological function[35]. It has also been shown that a single amino acid change in the CSD or the CTE of HP1α is sufficient to affect discrimination among its binding partners[36]. These observations demonstrate that subtle sequence variations between HP1α and HP1γ can be the source of major biochemical differences between the two paralogs and, ultimately, of distinct, even opposing, cellular functions. Amino acid variations can directly modify the secondary structure of HP1 or the CSD binding affinity to peptide. They may also explain why the different HP1 paralogs are subject to at least partially distinct post-translational modifications. These covalent processing events change the localization, function and interactome of the HP1 proteins and have therefore been proposed to constitute a subcode within the histone code[37]. In differentiating murine embryonic stem cells, HP1β phosphorylation at serine 89 is, for example, necessary for TRIM28 recruitment, heterochromatin-dependent silencing of pluripotency genes, and pluripotency exit[38]. Moreover, once phosphorylated at serine 83, HP1γ exhibits impaired silencing activity and strictly localizes within euchromatin, where it interacts with Ku70 and serves as a marker for transcription elongation[37]. The post-translational modifications may be induced downstream of the signaling pathways activated by environmental signals, providing a link between the cytoplasmic and nuclear events that regulate gene expression. Taken together, these observations suggest that the opposite roles of HP1α and HP1γ may result from subtle sequence differences that give each paralog the intrinsic ability to interact with distinct protein partners (*e.g.* transcription factors), and therefore to regulate different gene networks.

While most studies document concomitant inhibition of T cell proliferation and effector functions by Treg, the uncoupling we observed in HP1α-deficient CD4[+] T cells is not unprecedented. In a seminal paper, Chen and colleagues showed that Treg could inhibit the cytotoxic activity of tumor-specific CD8[+] T cells without affecting their activation, proliferation, homing and cytokine production[39]. Since then, similar observations have been made in other studies[40,41]. In a murine model of gastritis, it was, for example, reported that Treg repress the differentiation of autoreactive Tconv into Th1 cells without inhibiting their expansion in the draining lymph nodes[40]. In their system, as in ours, IFN-γ production by Tconv was thus inhibited by Treg without affecting their accumulation.

The field of cancer therapy has been revolutionized by the advent of immune checkpoint inhibitors (ICI). Since their initial approval in melanoma, these compounds have dramatically improved outcomes for cancer patients[42], including those suffering from hematologic malignancies[43]. However, ICI have limited efficacy and only a small percentage of patients respond to these therapies. Their partial failure can be explained, at least in part, by the persistence of a state of exhaustion in host T cells when the repression pathways have been short-circuited[44,45]. The immunosuppressive signals delivered within the tumor microenvironment indeed mobilize a molecular machinery, still largely uncharacterized, that strongly modifies the T cell epigenome and blocks access to the *cis*-regulatory elements that control effector genes expression[44]. In an attempt to overcome resistance to ICI, numerous clinical trials are currently testing the clinical benefit of combination treatment regimens consisting of ICI and chemotherapeutic agents or ICI-ICI combination[43,46]. Combining HP1α inhibition with peptidomimetic[47] could be a promising alternative to these strategies. In our bone marrow allograft model, we showed that HP1α-deficient CD4[+] T cells fail to efficiently repress their effector mechanisms when exposed to immunosuppressive signals, and our transcriptomic and epigenomic data support the notion that HP1α is necessary to shut down the genes involved in type 1 immunity. These observations suggest that HP1α inhibition could synergize with ICI treatment to reinvigorate exhausted T cells and promote tumor rejection. To avoid the side effects that could arise from the ubiquitous expression and pleiotropic functions of HP1 proteins, it might also be possible to inhibit HP1α expression not in vivo, but ex vivo to enhance the efficacy of CAR T cell-based therapies.

In mice, immune checkpoint receptors have been critically involved in both acquisition and maintenance of immunological self-tolerance, and immune checkpoints defects lead to severe immune dysfunction[48]. In human, single-nucleotides polymorphisms in the gene encoding PD-1 have been associated with increased susceptibility to a variety of autoimmune disorders, such as multiple sclerosis or rheumatoid arthritis[48,49]. Since we show here that HP1γ and SUV39H1 negatively regulate a set of genes critically involved in T cell anergy and exhaustion, including those encoding TOX, PD-1 and LAG-3, interfering with these molecules could be of interest to control allogeneic responses in transplant patients, but also to inhibit pathogenic immune responses in patients suffering from autoimmune pathologies. To trigger signaling through immune checkpoint receptors, various agonistic agents have been developed and are currently tested in clinical settings. Interestingly, our data suggest that inhibition of the

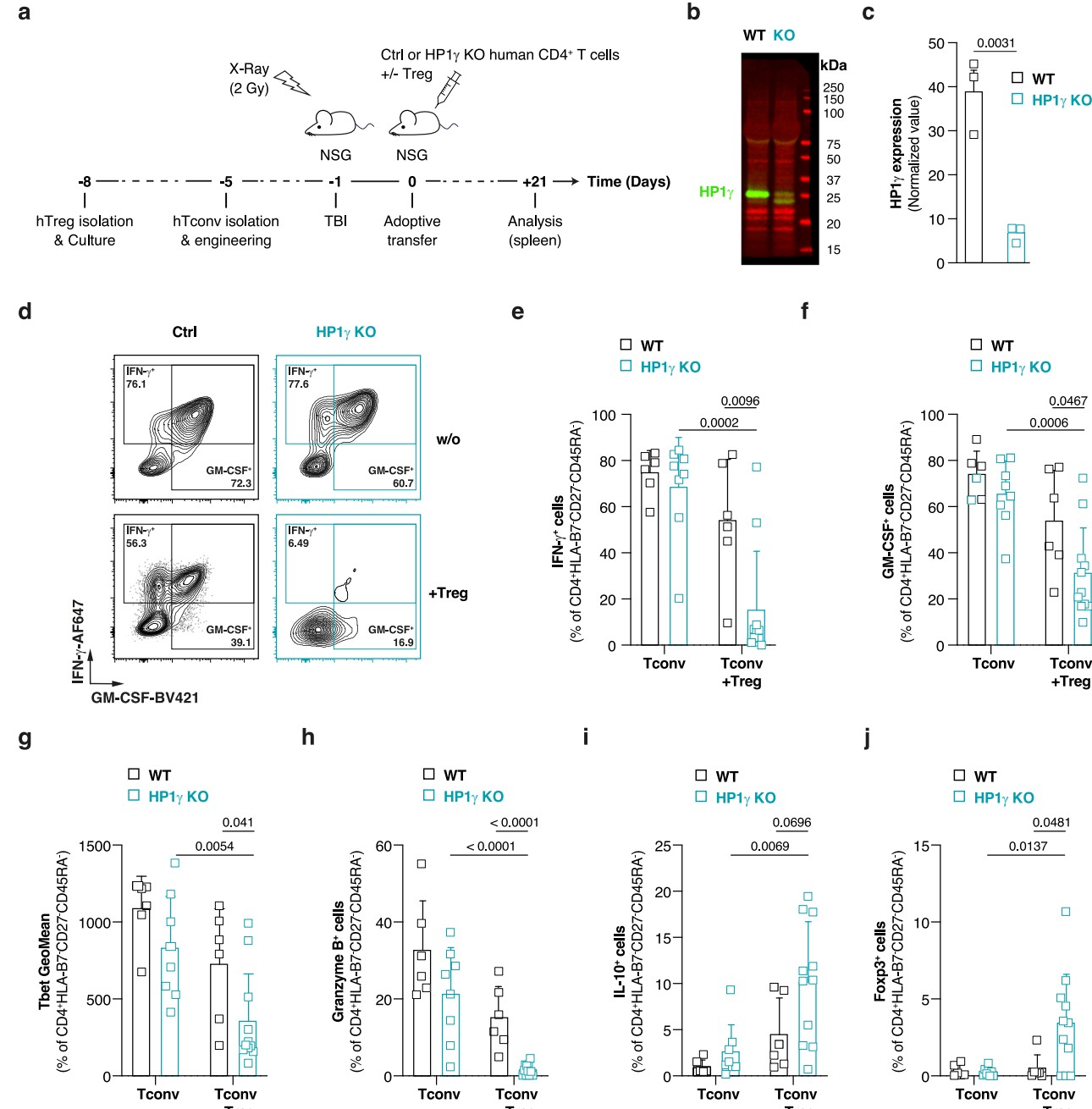

**Fig. 7 | HP1γ-deficient human CD4⁺ T cells are more susceptible to Treg-mediated suppression than wild-type cells.** HLA-B7⁻ WT or HP1γ-deficient human naive CD4⁺ T cells were injected *i.v.* into sublethally-irradiated NSG mice with or without ex vivo expanded HLA-B7⁺ human Treg. Three weeks after injection, splenocytes were isolated and the xenogeneic T cell response was analyzed by flow cytometry. **a** Experimental model. **b, c** Following genome editing by CRISPR-Cas9, the expression level of HP1γ was determined in WT and HP1γ KO human CD4⁺ T cells. A representative western-blot (**b**) and normalized expression levels for each experiment (**c**) are shown. **d** Representative dot-plots showing the percentage of HLA-B7⁻ WT and HP1γ KO human Tconv producing IFN-γ and GM-CSF. **e–g** Percentage of HLA-B7⁻ WT and HP1γ KO human Tconv producing IFN-γ (**e**), GM-CSF (**f**) or Granzyme B (**g**). **h** Expression level of T-bet in HLA-B7⁻ WT and HP1γ KO human CD4⁺ T cells. **i, j** Percentage of HLA-B7⁻ WT and HP1γ KO human CD4⁺ T cells producing IL-10 (**i**) or expressing Foxp3 (**j**). Data show values for biological replicates from two (**e–j**) or three (**c**) independent experiments. Horizontal bars represent mean ± SD (**e–j**) or mean ± SEM (**c**). *P* values were calculated using two-tailed unpaired t tests. Source data are provided in the Source data file.

SUV39H1-HP1 axis with compounds such as chaetocin, by boosting the expression of inhibitory coreceptors on the surface of T cells, could represent a synergistic treatment to these therapeutic strategies.

## Methods

The research carried out in this study complies with all relevant ethical and legal rules.

### Mice

Animals were used in accordance with a protocol reviewed and approved by the Institutional Animal Care and Use Committee of Region Midi-Pyrenees (France). The mutant mouse strains defective for HP1α, HP1β or HP1γ were established at the Mouse Clinical Institute (IGBMC, Strasbourg, France). Mice homozygous for a conditional ready *Hp1b* (*Hp1b*flox) or *Hp1g* (*Hp1g*flox) allele and carrying the

*Cd4*-cre transgene were crossed with mice only homozygous for the conditional ready allele to generate control and conditional HP1β-deficient (*Hp1b*[-/-]) or HP1γ-deficient (*Hp1g*[-/-]) mice within the same litter. Mice carrying a conditional ready *Hp1a* (*Hp1a*[flox]) allele were crossed with *CMV*-cre transgenic mice to excise the floxed sequence. Mice heterozygous for the mutated allele (*Hp1a*[+/-]) were then systematically intercrossed to generate control (*Hp1a*[+/+]) and HP1α-deficient (*Hp1a*[-/-]) mice within the same litter. SUV39H1-deficient mice were kindly provided by T. Jenuwein (Max Planck Institute, Freiburg, Germany)[50]. As the gene encoding SUV39h1 is carried on the X chromosome, *Suv39h1*[Y/+] males were crossed with heterozygous *Suv39h1*[+/-] females to obtain both *Suv39h1*[Y/+] and *Suv39h1*[Y/-] males within the same litter. These different lines were generated and maintained on a mixed (129/Sv x C57BL/6) genetic background. B6 *Foxp3*-Thy1.1 mice were provided by A Rudensky (Howard Hughes Medical Institute, Seattle)[51]. All mutant mouse strains as well as C57BL/6 J congenic mice expressing CD45.1 were bred under SOPF conditions at the Regional Center for Functional Exploration and Experimental Resources (CREFRE, INSERM UMS006, Toulouse). Experimental and control animals were co-housed with housing conditions using a 12 light/12 dark cycle, with a temperature between 20 and 24 °C with an average humidity rate between 40% and 70%.C57BL/6 J mice and B6D2F1 hybrids were obtained from Janvier Labs. Sex-matched 6- to 12-week-old wild-type and mutant littermates were used and compared in all experiments. All experiments involving animals were conducted according to animal study protocols approved by the local ethics committee (#16-U1043-JVM-496, 21-U1291-JVM-292, PI-U1043-JVM-20 and #32669-2021080215068127).

### Peripheral Blood Mononuclear Cells (PBMC)

The study was carried out on PBMC from volunteers who donated their blood and were recruited between June 2015 and September 2024 in the Occitania region in France. All donors reported feeling well and were evaluated to be in good general health by a physician at the French transfusion public service (Etablissement Français du Sang, EFS). Each blood donation was characterized for subject age, gender, blood type, CMV and T. gondii serostatus, and HLA-A2 expression. Blood was monitored for the absence of permanent contraindication for blood donation (HBV, HCV, HIV, or syphilis infection). This study on PBMC biobank from healthy adult subjects was approved by the French South-West & Overseas ethical committee and was registered at the French Ministry of Higher Education and Research (DC-2015-2488). Experiments were performed in agreement with the guidelines of the Declaration of Helsinki.

PBMC were isolated from buffy coats obtained 18 to 24 h after the blood collection and processed immediately upon receipt. PBMC were isolated by density-gradient sedimentation using Ficoll-Paque (Pancoll, Pan-Biotech). Isolated PBMC were frozen in heat-inactivated FCS (Gibco FBS, ThermoFisher) containing 10% tissue culture-grade DMSO (Sigma). Cryovials containing $20 \times 10^6$ PBMC were transferred at -80 °C at a cooling rate kept at 0.2–1 °C/minute using a cryopreservation module (StrataCooler Cryo Preservation Module, Agilent Technologies) and, after 24 to 96 h, transferred to liquid nitrogen. To ensure that assays could be properly interpreted, and to avoid confounding factors such as poor cryopreservation of the PBMC, we quantified cell viability after thawing and a $20 \pm 4$ h recovery period and ran the same quality controls on all PBMC included in this study. In average, mean (SD) x$10^6$ cells were recovered per cryovial with a mean cell viability of mean (SD) % after the exclusion of N subject PBMC that did not meet a $20 \pm 4$ h post-thaw viability criteria of $\geq 80\%$.

### Isolation of mouse and human T-cell subsets

Mouse Treg were isolated from the spleen of B6 *Foxp3*-Thy1.1 mice. For the in vitro experiments, Treg were enriched by positive selection on

MS column (Miltenyi Biotec) following staining with APC-conjugated anti-Thy1.1 antibody (OX-7, BD Biosciences) and incubation with anti-APC microbeads (Miltenyi Biotec). The resulting cell suspension was systematically enriched to over 95% in CD4+Thy1.1+ cells. For the in vivo experiments, the cell suspension underwent an additional selection step before culture. Treg, defined as CD4+CD25+Thy1.1+, were purified by fluorescence-activated cell sorting on a FACSAria Fusion cell sorter (BD Biosciences). The Treg population obtained was virtually pure after sorting and over 95% enriched after culture.

Control and mutant mouse Tconv were enriched from the spleen of control and mutant littermates, respectively. Splenic naive CD4+ T cells were obtained by negative selection using the mouse naive CD4+ T Cell isolation kit (Miltenyi Biotec) according to the manufacturer's instructions. The CD4+CD62L+CD44[-/low] T cell population was routinely more than 95% pure.

Human Treg were isolated from frozen PBMC using the EasySep Human CD4+CD127[low]CD25+ Regulatory T Cell Isolation Kit (Stemcell) according to the manufacturer's instructions. The CD4+CD127[low]CD25[high] Treg population was routinely more than 95% pure.

Human naive CD4+ T cells were isolated from frozen PBMC using the EasySep Human Naive CD4+ T Cell Isolation Kit II (Stemcell) according to the manufacturer's instructions. The CD3+CD4+CD45RA+ T cell population was routinely more than 95% pure.

### Generation of HP1-deficient human primary T cells

HP1γ-deficient human naive CD4+ T cells were generated using the Cas9/RNP transfection methods optimized by Seki et al[52]. Briefly, freshly isolated naive CD4+ T cells were resuspended in P2 nucleofection solution (Lonza), incubated with Cas9/RNP complex (Integrated DNA Technologies) and electroporated (EH100 pulse) using a 4D-Nucleofector (Lonza). Prior to injection, cells were eventually cultured for 3 to 5 days in resting conditions (complete medium (RPMI 1640 with 10% Fetal Calf Serum (FCS), 1 IU/mL penicillin, 1 µg/mL streptomycin, non-essential amino acids, 50 µM 2-Mercaptoethanol, 2 mM glutamine and 1 µM HEPES, all from ThermoFisher Scientific) supplemented with 5 ng/mL IL-7 (R&D Systems). crRNA guides targeting *Cbx3* (GCTTGAGCTGTAGGCGCGGA, GACCCGGAGCAGCTCGGAGG and GGCCTCCAACAAAACTACA) were provided by IDT.

### Western Blotting

Cell lysates were generated in RIPA buffer (Thermo Scientific) before being diluted in Protein Sample Loading Buffer (LI-COR) supplemented with 10 mM DTT. Proteins were separated by SDS-PAGE on 4–12% Bis-Tris gels (Thermo Fisher Scientific) before being transferred onto nitrocellulose membrane (Trans-Blot Turbo RTA Mini 0.2 µm Nitrocellulose Transfer Kit, Biorad). The membrane was treated with a total protein stain (REVERT Total protein Stain kit, LI-COR), and imaged in the 700 nm channel with an Odyssey imaging system. Immunoblotting was performed with antibodies specific for HP1α (#2616, Cell Signaling), HP1β (#8676, Cell Signaling) or HP1γ (2MOD-1G6, purified from B cell hybridoma supernatant) and IRDye 800CW-conjugated secondary antibodies (LI-COR) to detect the target in the 800 nm channel. Image Studio Software was used for total and target protein quantification. Uncropped blots are provided in the source data file and in the supplementary information file.

### T-cell cultures

Mouse Treg were cultured in complete medium supplemented with 500 IU/mL rhIL-2 (R&D Systems), 0.1 ng/mL rhTGF-β 1 (R&D Systems) and anti-CD3/CD28 coated Dynabeads (Thermo Fisher Scientific). After three days of culture, TCR stimulation was removed and cells were further cultured in fresh medium for 5 to 7 additional days. Cell viability was routinely higher than 95%.

To analyze Th cell differentiation in vitro, naive CD4+ T cells were cultured in 96-well flat bottom plates coated with 2 µg/mL anti-CD3ε

antibody (145-2C11, BioXcell) in a complete medium supplemented with 1 µg/mL anti-CD28 antibody (37.51, BioXcell). Unless otherwise stated, Th1 culture medium also contained 10 ng/mL rmIL-12 (R&D Systems) and 10 µg/mL anti-IL-4 antibody (11B11, BioXcell), and Th17 culture medium contained 10 ng/mL rmIL-1β (R&D Systems), 20 ng/mL rmIL-6 (R&D Sytems), 1 ng/mL rhTGF-β 1 and 10 µg/mL anti-IL-4 and anti-IFN-γ (XMG1.2, BioXcell) antibodies. After three days of culture, TCR stimulation was removed (except for Th17 cells) and cells were plated for three additional days in fresh conditioning medium supplemented with 10 ng/mL rmIL-23 (R&D Sytems) for Th17 cell cultures. When indicated, naive CD4$^+$ T cells were labeled prior to culture with 1 µM CellTrace Violet (Thermo Fisher Scientific).

To measure early T cell activation and immune checkpoints expression in vitro, naive CD4$^+$ T cells were cultured for two days in 96-well flat bottom plates coated with 2 µg/mL anti-CD3ε antibody in complete medium supplemented with 1 µg/mL anti-CD28 antibody. When indicated, 5 ng/mL Chaetocin (Sigma) or DMSO was added to the culture.

Prior to adoptive transfer into NSG hosts, human Treg were first cultured in ImmunoCult-XF T Cell Expansion Medium (Stemcell) containing ImmunoCult Human CD3/CD28/CD2 T Cell Activator (Stemcell), 500 IU/mL rhIL-2 and 100 ng/mL Rapamycin (Sigma). After three days of culture, TCR stimulation was removed and cells were further cultured in fresh medium for 3 to 5 additional days.

### T cell activation, proliferation and differentiation analysis by flow cytometry

To analyze transcription factor expression in mouse Th cells, lymphocytes were stained with the fixable viability dye eFluor 506 (Thermo Fisher Scientific) and labeled with PE-conjugated anti-T-Bet (4B10, Thermo Fisher Scientific), PE-coupled anti-RORγt (Q31-378, BD Biosciences) and eFluor660-conjugated anti-Foxp3 (FJK-16s, Thermo Fisher Scientific) antibodies by means of the Transcription Factor Staining Buffer Set (Thermo Fisher Scientific). For intracellular cytokine staining, cells were first stimulated with 10 ng/mL Phorbol 12-Myristate 13-Acetate (PMA, Sigma) and 1 µg/mL Ionomycin (Sigma) for 4 hours in the presence of 5 µg/mL Brefeldin A (BFA, Sigma). Cells were then labeled with the fixable viability dye eFluor 506 and stained with FITC-conjugated anti-IFN-γ (XMG1.2, BD Biosciences), PE-coupled anti-IL-2 (JES6-5H4, Thermo Fisher Scientific), PE-Cy7-conjugated anti-IL-17A (eBio17B7, Thermo Fisher Scientific) and APC-conjugated anti-GM-CSF antibodies (MP1-22E9, BD Biosciences) by using the Cytofix/Cytoperm™ Fixation/Permeabilization kit (BD Biosciences).

To analyze mouse T cell activation and expression of immune checkpoints, cells were labeled with the fixable viability dye eFluor 506 and stained with PE-conjugated anti-CD25 and PE-CF594-coupled anti-CD69 (H1.2F3, BD Biosciences) antibodies or with PE-Cy7-conjugated anti-PD-1 (J43, Thermo Fisher Scientific) and PE-coupled anti-LAG3 antibodies (C9B7W, BD Bioscience).

Data acquisition was performed on MACSQuant Analyzer 10 (Miltenyi Biotec) or LSRII (BD Biosciences) flow cytometer. Analyses were carried out using FlowJo software (BD).

### In vitro immunosuppression assay

Freshly purified naive CD4$^+$ T cells were labeled with 1 µM CellTrace Violet and then cultured in complete medium with X-ray irradiated (25 Gy, RS 2000, RadSource) CD45.1$^+$ splenocytes previously coated with anti-CD3ε antibody and with or without increasing number of freshly isolated Treg. After four days, cells were collected, incubated with anti-CD16/32 antibody (2.4G2, purified from hybridoma supernatant) and mouse and rat IgG, stained with the fixable viability dye eFluor 506 and then with APC-Cy7-coupled anti-CD45.1 (A20, BD Biosciences), PerCP-Cy5.5-coupled anti-CD45.2 (104, BD Biosciences), APC-conjugated anti-Thy1.1 and BUV395-labeled anti-CD4 antibodies.

Data acquisition was performed on LSRFortessa X-20 or LSRII flow cytometer. Analysis of CTV dilution within CD4$^+$CD45.2$^+$CD45.1$^-$Thy1.1$^-$ cells was carried out using FlowJo software.

### Mouse phenotyping

To analyze the T cell compartment in mutant mouse strains, spleen, thymus and mesenteric lymph nodes were collected and dissociated as previously described. The single-cell suspensions were then incubated with mouse and rat IgG and anti-CD16/32 antibody before being labeled with the fixable viability dye eFluor 506. Extracellular staining was performed with BV421-coupled anti-TCRβ (H57-597, BD Biosciences), BUV395-labeled anti-CD4, APC-Cy7-conjugated anti-CD8 (eBioH35-17.2, Thermo Fisher Scientific), PE-Cy7-conjugated anti-CD44 (IM7, BD Biosciences), APC-coupled anti-CD62L (MEL-14, Thermo Fisher Scientific) and PE-conjugated anti-CD25 antibodies. For transcription factor expression analysis, cells were labeled with FITC-coupled antibodies specific for Foxp3 using the Transcription Factor Staining Buffer Set (Thermo Fisher Scientific). To analyze cytokine production by memory T cell, cells were first stimulated with 10 ng/mL PMA and 1 µg/mL Ionomycin in the presence of 5 µg/mL BFA. Cells were then labeled with BV750-conjugated anti-IFN-γ, APC-R700-coupled anti-IL-2, BV605-labeled anti-IL-17A, FITC-conjugated anti-TNF (MP6 XT22, BD Biosciences) and APC-coupled anti-GM-CSF antibodies using the Cytofix/Cytoperm Fixation/Permeabilization kit (BD Biosciences). Data acquisition was performed on the MACSQuant Analyzer 10 flow cytometer. Analyses were carried out using FlowJo software.

### Bone marrow allograft

Bone marrow was isolated from C57BL/6 J (H2$^b$) and B6D2F1 (H2$^{bd}$) mice by centrifugation. The single-cell suspensions were then depleted from NK and T cells by complement-mediated lysis using rabbit complement (7,5%, Cedarlane) and antibody specific for Thy1.2 (AT83, hybridoma supernatant) and for the glycolipid asialo ganglioside-GM1 (Fujifilm Wako Chemicals). The single-cell suspensions were then injected intravenously into X-ray irradiated (split-dose, 13 Gy total) C57BL/6 J or CD45.1$^+$ B6 hosts. C57BL/6 J and B6D2F1 bone marrow cells were injected at a 1:1 ratio in the presence of freshly isolated wild-type or mutant naive CD4$^+$ T cells and with or without ex vivo expanded Treg at various ratios. To control engraftment in each experiment, a group of mice was also injected only with the bone marrow cell mixture. Bone marrow engraftment was analyzed on blood samples using FITC-conjugated anti-H-2K$^b$ (AF6-88.5, BD Biosciences), PE-coupled anti-H-2K$^d$ (SF1-1.1, BD Biosciences), Pacific Blue-labeled anti-CD19 (1D3, BD Biosciences) and APC-eFluor780-conjugated anti-Ly6G (RB6-8C5, Thermo Fisher Scientific) antibodies, and analyzed by flow cytometry.

### Ex vivo analysis of CD4$^+$ T cells from transplanted host

Splenocytes were stained with the fixable viability dye eFluor 506 and then labeled with APC-Cy7-conjugated anti-CD45.1; BV421-coupled anti-CD45.2, PE-associated anti-H-2K$^d$, APC-conjugated anti-Thy1.1, V500-labeled anti-CD4, BV711-coupled anti-FR4 (12A5, BD Biosciences), BV605-conjugated anti-CD73 (TY/11.8, BioLegend) and BB700-associated anti-TCR Vβ6 (RR4-7, Thermo Fisher Scientific). For transcription factor expression analysis, cells were labeled with PE-Cy7-conjugated anti-Foxp3 antibodies using the Transcription Factor Staining Buffer Set (Thermo Fisher Scientific). For cytokine production analysis, cells were previously stimulated with 10 ng/mL PMA (Sigma) and 1 µg/mL Ionomycin (Sigma) for 4 h in the presence of 5 µg/mL BFA (Sigma), and then stained with BV750-coupled anti-IFN-γ, APC-R700-associated anti-IL-2 and BV605-conjugated anti-IL-17A antibodies using the Cytofix/Cytoperm Fixation/Permeabilization kit (BD Biosciences). Cells were analyzed by flow cytometry as described above.

## xenogeneic Graft-Versus-Host Disease (xGVHD)

Control or HP1γ-deficient human naive T cells (5×10⁵) were injected *i.v.* into X-ray irradiated (2 Gy) NSG mice with or without ex vivo expanded human Treg (10⁶). Three weeks after injection, splenocytes were isolated, incubated with mouse and rat IgG and anti-CD16/32 antibody (2.4G2) and labeled with the fixable viability dye eFluor 506. Extracellular staining was performed with APC-Cy7-conjugated anti-CD3 (OKT3, BioLegend), BUV395-coupled anti-CD4 (SK3, BD Biosciences), BUV605-conjugated anti-HLA-B7 (BB7.1, BD), AF700-labeled anti-CD45RA (HI100, BioLegend) and FITC-labeled anti-CD27 (LG.7F9, Thermo Fisher Scientific) antibodies. For transcription factor expression analysis, cells were labeled with PE-conjugated anti-T-Bet and APC-coupled anti-Foxp3 (236 A/E7, Thermo Fisher Scientific) antibodies using the Transcription Factor Staining Buffer Set (Thermo Fisher Scientific). For cytokine production analysis, cells were previously stimulated with PMA and Ionomycin in the presence of BFA, and then stained with AF647-coupled anti-IFNγ, BV421-coupled anti-GM-CSF, PE-labeled anti-Granzyme B (GB11, BD Biosciences), BV750-associated anti-IL-2 (MQ1-17H12, BD Biosciences) and BB700-conjugated anti-IL-10 (JES3-19F, BD Biosciences) antibodies using the Cytofix/Cytoperm Fixation/Permeabilization kit. Data acquisition was performed on Symphony A5 (BD Biosciences) and analyses were carried out using FlowJo software.

## RNA-Seq

To analyze Tconv and Treg from transplanted mice, CD4⁺ T cells were first enriched by positive selection using mouse CD4 (L3T4) microbeads (Miltenyi Biotec) and then stained with APC-Cy7-conjugated anti-CD45.1, BV421-coupled anti-CD45.2, FITC-conjugated anti-H-2K^b, PE-labeled anti-H-2K^d, APC-coupled anti-Thy1.1, V500-conjugated anti-CD4 and BB700-labeled anti-TCR Vβ6 antibodies. Treg, defined as CD4⁺CD45.1⁻CD45.2⁺Thy1.1⁺H-2K^b⁺H-2K^d⁻, and donor-specific Tconv, defined as CD4⁺CD45.1⁻CD45.2⁺Thy1.1⁻H-2K^b⁺H-2K^d⁻TCRVβ6⁺, were eventually purified by fluorescence-activated cell sorting (FACSAria Sorp or Fusion, BD Biosciences). To analyze memory T cells from mutant and control littermates, CD4⁺ T cells were enriched from splenocytes using the Dynabeads Untouched Mouse CD4 Cells Kit (Thermo Fisher Scientific) and then stained with Pacific Blue-conjugated anti-CD4, PE-Cy7-coupled anti-CD44, APC-labeled anti-CD62L and PE-coupled anti-CD25 antibodies. Memory T cells, defined as CD4⁺CD25^low/-CD62L⁻CD44^high were eventually purified by FACS (FACSAria Sorp or Fusion, BD Biosciences).

The RNA libraries were prepared from purified RNA (RNeasy Micro Kit, Qiagen) using the SMART-Seq® v4 PLUS kit (Takara Bio) and the SMARTer RNA Unique Dual Index kits (for naive CD4⁺ T cells and Tconv isolated from transplanted mice), or using the TruSeq® Stranded Total RNA Library Prep Gold (Illumina) and IDT for Illumina TruSeq UD Indexes (for Treg) or using the Illumina Stranded mRNA Prep Ligation Kit and Illumina RNA UD Indexes (for memory T cells). Samples were indexed and sequenced (paired-end reads of 150 bp) on the NovaSeq 6000 system (Illumina).

Raw sequencing reads were processed using the nf-core/RNA-seq pipeline v.3.9 developed using Nextflow[53,54]. Briefly, this pipeline trims and removes low-quality reads using Cutadapt v.3.4 with the wrapper Trim Galore! v.0.6.7[55,56]. It then aligns reads to the Ensembl GRCm39 genome and annotation v.107 using STAR v.2.7.10a[57]. Finally, gene expression is quantified with Salmon v.1.5.2[58]. This pipeline also generates quality controls by running FastQC v.0.11.7[59] and Qualimap v.2.2.2[60]. The sample HP1g-S12 was excluded from the analysis because its per sequence GC content, percentage of reads mapped to exonic regions and 5'-3' bias were abnormal.

## ATAC-Seq

ATAC-Seq was performed on FACS-sorted donor-specific CD4⁺ T cell as previously described[6] with some modifications. Briefly, 50,000 cells were lysed in ice-cold lysis buffer and the transposition reaction was performed using the Tn5 transposase (ILMN Tagment DNA Enzyme and Buffer, Illumina) at 37 °C for 30 min. DNA was purified using the Qiagen MinElute kit (Qiagen) and the libraries were prepared using the NEBNext High-Fidelity 2X PCR Master Mix (New England Biolabs) and indexes (Sigma) compatible with Illumina sequencing technology. The libraries were then purified twice using AMPure XP beads (Beckman) following a double-sided protocol to remove primer dimers and large fragments. Samples were multiplexed and sequenced (100 bp, paired-end reads) on the NovaSeq 6000 system (Illumina) from the Genomic and Transcriptomic Platform of the GenoToul (Toulouse, France). Raw sequencing reads were trimmed and low-quality reads were removed using Trim Galore! v.0.6.7[56]. Trimmed reads were then aligned to the Ensembl GRCm39 genome with BWA-MEM 2 v.2.2[61]. Mapped reads were filtered using Samtools v.1.9[62] to keep only reads that were paired in primary alignments with a minimum quality of 30 and not in the blacklist regions. To define these blacklist regions, we downloaded the mm10 blacklist coordinates of ENCODE[63] and we converted them into GRCm39 coordinates using the LiftOver tool of the UCSC Genome Browser. After filtering, duplicated reads were detected and removed by MarkDuplicates from the Picard tools v.2.20.7. Deduplicated and filtered alignments were given to HMMRATAC v.1.2.10 for peak detection[64]. Peaks detected for each sample were merged together with BEDTools 2 v.2.29.0[65] and consensus peaks were quantified for each sample using featureCounts from subread v.2.0.3 with parameters -F SAF -O –fracOverlap 0.2 -p –donotsort[66]. Finally, consensus peaks were annotated with annotatePeaks.pl from HOMER v.4.10.4[67] and GRCm39 annotation v.107. This tool assigns a peak to the nearest gene. Unless stated otherwise, tools were run using default parameters.

When calculating metrics to detect low-quality samples, we observed that HP1a-S33, HP1a-S35, and HP1a-S36 samples had a fraction of reads in peaks below 0.15 and a TSS enrichment value below 10. We therefore decided to exclude them before running further analyses.

## Bioinformatic analysis

**RNA-seq.** We used the R package DESeq2 v.1.38.3 to normalize gene expression and perform differential expression analyses[68]. We identified differentially expressed genes (DEG) as genes with an absolute $log_2$ fold change greater than 1 and an adjusted p-value below 0.05.

**ATACseq.** Normalization of consensus peak signal and differential accessibility analyses were performed with the R package edgeR v.3.40.2[69–71]. Differentially accessible peaks were defined as peaks with absolute log2 fold change greater than 1 and adjusted p-value below 0.05.

**Correlated modules definition.** Modules of highly correlated genes were created using the Weighted correlation network analysis (WGCNA) R package v.1.72-1[72]. To do so, we selected differentially expressed genes and ran multiple WGCNA. The analyses were run on genes overexpressed in HP1γ KO Tconv exposed to Treg compared with wild-type Tconv exposed to Treg. The expression matrix of these genes in each sample was given to WGCNA after choosing the soft-thresholding power. Then, WGCNA calculates the co-expression adjacency and transforms it into a Topological Overlap Matrix. It finally defines clusters using hierarchical clustering with the average method. Modules with very similar expression profiles are then merged. WGCNA identified 4 clusters for genes overexpressed in HP1γ KO Tconv compared with wild-type Tconv exposed to Treg with 51 (cluster 1), 340 (cluster 2), 58 (cluster 3) and 20 (cluster 4) genes, respectively.

**scRNAseq analyses.** TPM counts per cell and the correspondence between cells and defined clusters were loaded in a Seurat object using

the Seurat R package v.5.0.1[73]. From the TPM counts we calculated the 4000 most variable features. We then scaled the data and performed the dimensionality reductions: PCA, UMAP from the 30 first PCA. For subsequent analyses, we only selected cells from clusters 5, 8, 10 and 11, i.e. 7232 cells. Those cells were reclustered with a resolution of 0.6. We selected only clusters with CD4$^+$ T cells. To avoid contamination by CD8$^+$ T cells, we also filtered out cells with more than 1 TPM of CD8A or CD8B. We reclustered CD4$^+$ T cells with a resolution of 0.4 and we defined cell types based on marker genes of each cluster. Finally, we calculated the DEG between Teff and Tex with the FindMarkers function with parameter logfc.threshold = 0 to keep all the genes.

**Enrichment analyses.** Over Representation Analyses (ORA) and Gene Set Enrichment Analyses (GSEA) were performed using clusterProfiler R package v.4.6.2[74,75]. Enrichments were calculated for GO database, KEGG database and gene sets manually curated (Supplementary Data 3 and 4) and DEG from our own datasets. ORA were performed on a specified gene list with the background set to all expressed genes. GSEA were performed on genes sorted by log$_2$ fold change for bulk RNAseq. For Teff vs Tex scRNAseq analyses, genes were sorted on average log2 Fold Change and DEG from our own datasets were previously converted to human homologous genes thanks to the MGI v.6.13[76] database.

**Statistics & Reproducibility**
Statistical parameters, including the exact value and significance of n and precision measures (Mean or Median ± SEM or SD), as well as statistical significance are reported in the figures and figure legends. No statistical method was used to predetermine the sample size. As indicated above, the RNA-seq sample HP1g-S12 and the ATAC-seq samples HP1a-S33, HP1a-S35 and HP1a-S36 were excluded from the analyses due to poor quality. No other data were excluded from the analyses. Independent experiments were performed at least three times to allow statistical analysis (with the exception of the xGvHD experiments which were carried out twice). The main flow cytometry gating strategies are provided in the Supplementary Information.

**Reporting summary**
Further information on research design is available in the Nature Portfolio Reporting Summary linked to this article.

## Data availability
Raw and processed data files from ATAC-seq and RNA-seq experiments have been deposited in the NCBI Gene Expression Omnibus (http://www.ncbi.nlm.nih.gov/geo/) under accession number GSE246831. The scRNAseq dataset used in this study has initially been published by Sade-Feldman and collaborators[28]. It is available in the NCBI Gene Omnibus Expression database under the accession number GSE120575. Source data of the main and Supplementary figures are provided in the Source data file. Source data are provided with this paper.

## Code availability
The code used to perform the bioinformatics analyses is available through our Code Ocean capsules 4177274 and 5889808.

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

## Acknowledgements

We thank T. Jenuwein (Max Planck Institute, Freiburg, Germany) for providing *Suv39h1*<sup>-/-</sup> mice. We also acknowledge F.-E. L'Faqihi-Olive, V. Duplan-Eche, A.-L. Iscache and H. Garnier for technical assistance at the flow cytometry facility of Infinity, the personnel of the US006 ANEXPLO/CREFRE animal facility for expert animal care, D. Rozet for administrative assistance, the GeT and Bioinformatics platforms from the Genotoul (Toulouse, Région Occitanie, France), the Genomic and Transcriptomic platform of the CRCT (Toulouse, France) and C. Delmas at the Technology Cluster of the CRCT. We thank R. Romieu-Mourez and P-E. Paulet from the human immune-monitoring platform at Inserm UMR 1291 (Infinity) for the supervision of the PBMC biobank from healthy adult subjects. We are grateful to members of the "Integrative T cell Immunobiology" laboratory for advice, discussions and technical help. We also thank Dr Rod Bremner (Lunenfeld Tanenbaum Research Institute, Mt Sinai Hospital, Toronto, Canada) for careful reading of the manuscript. This work was supported by Agence Nationale de la Recherche (ANR, ANR-19-CE15-0022 "ImmuneTrans" (O.J.) and ANR-20-CE15-0005 "MoRegPaF" (O.J.)), Fondation pour la Recherche Médicale (FRM, AJE201212 and EQU202203014703 (O.J.)) and Association pour la Recherche sur le Cancer (ARC, ARCPJA2021060003837 (O.J.)).

## Author contributions

Contributions: J.N. designed the project, carried out experimental work and contributed to writing the manuscript. K.L. and M.Z. are co-second authors. K.L. critically contributed to carry out the experimental work. M.Z. performed the bioinformatic analysis of NGS data and contributed to writing the manuscript. C. Detraves, A.C., A.S., R.T., C. Demont and V.A. carried out experimental work. C.J. and F.C. provided critical materials and contributed to writing the manuscript. J.P.M.v.M. contributed to writing the manuscript. O.P.J. conceived the project, supervised the study and wrote the manuscript.

## Competing interests

O.P.J., J.N., K.L., and M.Z. have previously filed a patent application based on the use of HP1 to modulate Th cell-dependent immunity. The remaining authors declare no conflict of interest.
