## [Transparent Peer Review file · Nature Communications]

Heterochromatic gene silencing controls CD4⁺ T cell susceptibility to regulatory T cell-mediated suppression in a murine allograft model

Corresponding Author: Dr Olivier Joffre

Version 0:

Reviewer comments:

Reviewer #1

(Remarks to the Author)

In the manuscript entitled "Heterochromatic gene silencing controls CD4 T cell susceptibility to regulatory T cell-mediated suppression" Noguerol et al. examine the crosstalk between conventional and regulatory T cells at the chromatin level. The authors use a murine model for bone marrow transplantation in which lethally irradiated hosts are reconstituted with a 1:1 mixture of syngeneic and semi-allogeneic bone marrow, and graft rejection is mediated through CD4 T cells and inhibited by Tregs. Treg-mediated T cell suppression involves HP1 α -dependent silencing of the Th1 and Th17 gene networks while HP1 γ acts as a negative regulator of T cell exhaustion. Collectively, the authors nicely demonstrate that the heterochromatin regulators HP1 α and HP1 γ act in opposite roles to regulate the dialogue between conventional and regulatory T cells. The manuscript is clearly written, and the main points are clearly stated and supported. Below are a few questions the authors could address to strengthen this work.

Specific Comments:

- 1). The authors show some very important and likely clinically relevant points pertaining to allogeneic immune responses. While the manuscript focuses on murine model systems, it would be really nice to show this in patient samples. Are there any human-based datasets that the authors could mine to further support their claims?
- 2). Are the HP1 α and HP1 γ -mediated heterochromatic changes stable over time or readily reversible? How would this impact on therapeutic applicability?
- 3). The authors nicely describe HP1 α and HP1 γ as paralogs that have the same structure and functional domains and undergo homo- or heterodimerization with binding partners that contain a PXVXL motif. Do the authors have any further insight into the sequence differences or variations in transcription factor interaction that could account for the opposite role of HP1 α and HP1 γ ?

(Remarks on code availability)

Reviewer #2

(Remarks to the Author)

The authors have examined the role of three related chromodomain proteins that can mediate epigenetic changes in CD4⁺ and CD8⁺ T lymphocytes. This report claims a specific effect of two of these proteins HP1a and HP1g on the function of T regulatory cells in vivo in a complex model of Treg mediated acceptance of allogeneic bone marrow transplantation. The results suggest that HP1a controls the capacity of Treg to mediate suppression of effector cytokine production so that in its absence cytokine production (Th1 and TH17) by T effector cells is not suppressed and allogeneic bone marrow is rejected. Conversely, HP1g has the opposite effect and in its absence results in an effector population that is easier to suppress with enhanced acceptance of an allogeneic transplant. This is a very novel concept, but there are numerous technical and theoretical problems with this report that weaken the conclusions drawn by the authors:

1. The mouse strains used as the effector populations are not genetically identical. Wild type B6 CD4⁺ T cells are used as

the control populations while the three knock out strains are maintained on a mixed 129/B6 background. This may introduce unknown variations in the graft rejection studies. Furthermore, one of the knock out strains is a global knock out while the other two (HP1b and HP1c) are T cell conditional knock outs. Differential effects on T cell development may result in these different strains.

2. While the paper and previous studies by this group have examined the effects of Treg on the control of allogeneic graft rejection, the studies in the present report also involve the response to the superantigen mmtc-7 so this is not a pure allo response. More importantly, in many of the figures the authors restrict their analysis to the T cell subset (Vb6) which recognizes the superantigen. It would have been preferable to illustrate the global anti-allo response. In particular, the capacity of superantigens to induce pTreg or so called anergic cells may not be representative of the the typical response to alloantigens.

3. The major problem with the paper is the characterization of the effector cell populations used to induce graft rejection. While the authors claim that CD4 cell differentiation in the steady state is completely normal in the knock out strains, their analysis is rather superficial. One potential explanation is that the effector cells from the HP1a^{-/-} mice may contain a higher proportion of memory or primed T cells and are harder to suppress, while the effector cells from the HP1g mice have fewer memory cells and are easier to suppress. The authors claim to have used naive T cells in all their studies, but they need to document that the percentages of these populations are equivalent to the wild type control. T cell suppression in the graft rejection model is subtle and the authors claim in previous papers can only be seen with Treg populations that have been expanded in vitro as in the present paper.

4. The claim that the HP1a effector cells are only resistant to suppression of cytokine production but not expansion in vivo is somewhat unusual as most studies of Treg suppression in vivo show marked suppression of effector cell expansion in addition to cytokine production.

5. The fate of the transferred Treg in these different combinations of effector cell mice is not examined.

6. It is also surprising that the RNA-seq studies on the HP1 deficient strains did not reveal any abnormalities, but the authors limited their studies to naive cells and not memory populations generated in vivo. why are abnormalities in cytokine gene expression only seen after exposure to Treg in vivo?

7. No data is presented to support the claim that the anergic cells generated from HP1g^{-/-} effector cells are in fact exhausted. Since they express Vb6, they could easily be restimulated with MMtv7 expressing cells.

8. While PD-1 is a marker of exhausted cells, it is also a marker of recently activated cells. The exhausted phenotype should be tested again with MMtv stimulation.

9. One more theoretical issue is how do the Treg mediate suppression--directly by contact with the T effector cells or indirectly via acting on APC (as is widely believed by some in the field)? What triggers the activity of members of the HP1 family? TCR signals? Costimulatory signals?

(Remarks on code availability)

Reviewer #3

(Remarks to the Author)

The manuscript by Joffre and colleagues focuses on the well-known phenomenon that the activation of effector T cells can be controlled by Tregs. Although several mechanisms are known by which Tregs exert the suppression, knowledge of signaling and consequences in the target cells is incomplete. The submitted manuscript describes a link between the Treg-mediated suppression and the heterochromatin proteins alpha and gamma. Using a mouse model of bone marrow transplantation, the authors show that HP1a-deficient cells fail to repress their effector mechanisms and that HP1g-deficient cells are more efficient in suppressing effector T cells. In addition, the authors prepared T cells from both knockout scenarios subjected them to ATAC-seq and RNA-seq and identified key pathways/genes for the observed effects.

Overall, I recommend the manuscript for publication as it is sufficiently novel and of interest to a wider group of scientists. However, a few points need to be clarified and improved before the paper can be finally approved.

Major points

The statistical method in Figure 1C, 2B-D, 3G, 4B,D, 5B-F, 6B, I, K is unclear.

The effect shown in Figure 2E was interpreted by an unaltered Treg-induced contraction. The gating of the dot plot and the explanation is not completely clear to me. Could the authors exclude that the CD45.2+ T cell reduction is just a consequence of filling the T cell niche in the presence of another population in the spleen? Is it competition or Treg-induced contraction in the used model? Please provide more data for your interpretation.

The authors identified 54 genes that were more accessible and expressed in HP1a Ko cells compared to WT cells. However, these genes are not listed despite the presentation of pathway analyses in Figure 3. It would be beneficial to include a list of the genes with associated values or as a heatmap (might be reduced to top hits). The authors should consider moving some of the moderately informative data, such as Figure 3I, to the supplement. The same applies to the similar analysis of the HP1g Ko cells. Figure 6E lists only a few genes, with no information about the remaining genes and their values. The genes from both experiments should appear in the discussion.

Tregs were expanded before transfer. Did the authors purify the Tregs after culture? What was the purity of the transferred Tregs? This should be mentioned in Material and Methods. Please provide a quality check of the Tregs by flow cytometry in the Supplement.

Minor points

Please introduce TCR Vb6 earlier. Explanation is in Line 189-190, but appears already at the beginning of this result paragraph.

Figure 6A: Title for color range is missing.

Figure S4 I,K: Please add a title for the organ.

Figure 6A: Which comparison was used to define the clusters? Tconv group? Please ensure that it is clearly stated in the text.

The wording "..., or not, ..." in the figure legends is a bit unusual and difficult to understand. It is recommend to revise the legends and use easy-to-understand descriptions.

Please also revise the methods:

1. Some paragraphs can be shortened.
2. Provide exact information for antibody fluorochromes (e.g. Line 544).
3. Were the Tregs isolated by two different methods? MACS or MACS and FACS? Please, indicate the method used for each type of experiment.
4. Line 706: Is Nextflowewel a typo?

(Remarks on code availability)

Reviewer #4

(Remarks to the Author)

This study meticulously investigates the role of HP1 α and HP1 γ in regulating T cell responses, particularly focusing on their impact on the interaction between Tconv and Tregs. The authors employed a well-characterised mouse model of bone marrow transplantation to elucidate the molecular mechanisms underlying the crosstalk between Tconv and Treg, revealing novel insights into the heterochromatin-dependent regulation of T cell sensitivity to immunosuppressive signals.

The findings demonstrate that Tconv deficient in HP1 α exhibit reduced sensitivity to Treg-mediated suppression, leading to inefficient control of allogeneic responses. This observation is further corroborated by the resistance of HP1 α -deficient Th1 and Th17 cells to Treg-mediated inhibition, suggesting a critical role of HP1 α in modulating T cell responses through chromatin remodelling. The comprehensive analysis, including RNA-seq and ATAC-seq, provides compelling evidence that HP1 α deficiency leads to an upregulation of genes associated with T cell activation and Th17 cell-dependent immunity, thus highlighting the gene networks regulated by HP1 α in the context of Treg-mediated suppression.

In contrast, the study also delves into the role of HP1 γ , presenting intriguing findings that HP1 γ deficiency enhances Treg's ability to suppress allogeneic T-cell responses. This effect is attributed to the modulation of a gene network associated with T cell exhaustion and anergy by HP1 γ , as evidenced by the differential expression and accessibility of genes involved in immune checkpoints and immunosuppressive cytokine production in HP1 γ -deficient cells.

The experiments are well-designed and include enough details. However, there are some questions and comments as below:

1. Model System Relevance: The study predominantly relies on a mouse model of bone marrow transplantation to elucidate the role of HP1 α and HP1 γ . While this model is invaluable for dissecting molecular mechanisms, the direct translatability of findings to human biology and disease contexts may be limited. Differences between mouse and human immune systems could affect the applicability of these results in clinical settings.
2. Lack of Human Data: The absence of validation in human T cells or patient samples limits the study's immediate relevance to human health and disease. Incorporating experiments using human cells could help verify the conservation of observed mechanisms across species and enhance the study's clinical significance. Did the authors test any samples from patients with autoimmunity or cancer, for instance?
3. Long-term Effects and Immune Tolerance: The study focuses on relatively short-term outcomes following bone marrow transplantation and T cell transfer. The long-term effects of HP1 α and HP1 γ deficiency on immune tolerance, autoimmunity, and memory T cell responses are not explored. These aspects are crucial for understanding the full implications of HP1 α/γ regulation in immune homeostasis and disease. This question has clinical implications, for instance, the potential role of the identified mechanisms in longer-term conditions such as chronic GvHD.
4. Mechanistic Insights into HP1 α/γ Function: While the study provides significant insights into the consequences of HP1 α and HP1 γ deficiency on T cell function and regulation, the precise molecular mechanisms through which these proteins exert their effects remain partially elucidated. Also, a crucial aspect, for instance, is whether this change of function is cell contact-dependent. Can you see the same effect in a trans-well culture?

A few specific questions:

-Page 7: line 171: Authors say, "in the absence of Tregs...." I am not sure why the presence or absence of Tregs is relevant to the phenocopy of wild-type cells? Could you please explain?

-Page 9, line 270: "When cultured under Th1- or Th17-polarizing conditions, wild-type and HP1 γ -deficient T cells also expressed similar levels of the lineage-determining master regulators T-bet and ROR γ t" Did you do the same for HP1 α ?

-Page 11: Did you try to block PD-1 and LAG3 on HP1 γ -deficient T cells and compare the function?

-Page 14: "Interestingly, our data suggest that inhibition of the SUV39H1-HP1 axis with compounds such as chaetocin, by boosting the expression of inhibitory coreceptors on the surface of T cells, could represent a synergistic treatment to these therapeutic strategies." I agree, but did you try to test this in an in-vitro model?

Minor comment:

Several complex sentences take a few readings to understand and need simplification. For instance:

"In contrast to their wild-type counterparts, HP1 α -deficient cells fail to efficiently repress their effector mechanisms when exposed to Treg, while HP1 γ -deficient cells are more sensitive to Treg-mediated suppression."

(Remarks on code availability)

Version 1:

Reviewer comments:

Reviewer #1

(Remarks to the Author)

The authors have addressed all of this reviewer's comments. I recommend this manuscript for publication.

(Remarks on code availability)

Reviewer #2

(Remarks to the Author)

None

(Remarks on code availability)

Reviewer #3

(Remarks to the Author)

In the revised manuscript from Nogueroles et al. my main points for a revision were the lack of statistical methods in the experiments shown, the lack of clarity about the Treg contraction shown in Figure 2, the lack of gene tables in Figure 3 and the unclear methodological information about the purification of the cell population used. These major points and also the minor points have been addressed. I am satisfied with the authors' response and the new information provided in the revised manuscript will help to better understand the experiments described. In addition, the new data sets provided to answer the questions of other reviewers have also improved the quality of the paper and helped to strengthen the authors' interpretation.

There is one small point that still needs to be corrected: It would be nice if the authors could provide references for their new statement in lines 99-100.

(Remarks on code availability)

Reviewer #4

(Remarks to the Author)

The authors have addressed my comments and questions adequately.

(Remarks on code availability)

Reviewer #1 (Remarks to the Author):

In the manuscript entitled “Heterochromatic gene silencing controls CD4 T cell susceptibility to regulatory T cell-mediated suppression” Noguerol et al. examine the crosstalk between conventional and regulatory T cells at the chromatin level. The authors use a murine model for bone marrow transplantation in which lethally irradiated hosts are reconstituted with a 1:1 mixture of syngeneic and semi-allogeneic bone marrow, and graft rejection is mediated through CD4 T cells and inhibited by Treg. Treg-mediated T cell suppression involves HP1 α -dependent silencing of the Th1 and Th17 gene networks while HP1 γ acts as a negative regulator of T cell exhaustion. Collectively, the authors nicely demonstrate that the heterochromatin regulators HP1 α and HP1 γ act in opposite roles to regulate the dialogue between conventional and regulatory T cells. The manuscript is clearly written, and the main points are clearly stated and supported. Below are a few questions the authors could address to strengthen this work.

Specific Comments:

1) The authors show some very important and likely clinically relevant points pertaining to allogeneic immune responses. While the manuscript focuses on murine model systems, it would be really nice to show this in patient samples. Are there any human-based datasets that the authors could mine to further support their claims?

We thank the reviewer for raising this very important point. We used two complementary approaches to begin testing whether our observations could be clinically relevant:

- To assess whether targeting HP1 α could be of interest in cancer patients, we first compared the transcriptomic signature induced by HP1 α -deletion in murine CD4⁺ T cells to the transcriptome of tumor-infiltrating T lymphocytes (TILs) isolated from patients with advanced melanoma (**new Figures 3K-3N**). Interestingly, we observed a significant enrichment of genes downregulated in HP1 α -deficient cells among those that define human exhausted T cell identity (**Figure 3N**). These data suggest that inhibiting HP1 α in tumor-specific human CD4⁺ T cells could, as it does in allospecific murine Tconv (**Figures 3C and 3D**), repress the establishment of the exhaustion program and thus promote a protective immune response.
- To test whether HP1 γ -deficiency could sensitize human CD4⁺ T cells to immunosuppressive signals, we set up a mouse model of xenogeneic graft-versus-host-disease (xGvHD) in which immunodeficient NSG hosts were injected with human naive CD4⁺ T cells alone or in the presence of human Treg (**new Figure 7A**). In this system, and in the absence of suppressive cells, a strong xenogeneic Th1 response developed within weeks of injection (**new Figures 7D-7H**). At a Treg:Tconv ratio of only 2:1, the co-injection of suppressor cells did not significantly inhibit xenospecific wild-type T cells. In contrast, under the same conditions, Treg effectively repressed the Th1 response mediated by human naive CD4⁺ T cells in which HP1 γ -expression had previously been inactivated by CRISPR-Cas9 (**new Figures 7D-7H**). Interestingly, some rare but detectable mutant cells also produced the anti-inflammatory cytokine IL-10 and expressed the transcription factor Foxp3 when exposed to Treg (**Figures 7I and 7J**). Taken together, these *in vivo* data suggest that HP1 γ regulates the susceptibility of human CD4⁺ T cells to immunosuppressive signals, and that targeting this molecule could be clinically relevant for repressing pathogenic T cell responses in transplanted individuals or in patients suffering from autoimmunity.

These data therefore support the idea that the HP1-dependent chromatin mechanisms we identified in mice could also be at work in human T cells.

2) Are the HP1 α and HP1 γ -mediated heterochromatic changes stable over time or readily reversible? How would this impact on therapeutic applicability?

We thank the reviewer for his/her very interesting question. HP1-dependent gene silencing mechanisms have been described as stable and inheritable through multiple cell divisions.

Homodimerisation of HP1s *via* their chromoshadow domain results in the formation of a dense protein network that locks up chromatin in a highly compacted state (Canzio D et al., Nature 2013). In differentiating Th cells, we and others already showed that HP1s and partners (*i.e.* SETDB1, SUV39H1) are critically involved in the repression of genes associated with alternative fates (Allan RS et al., Nature 2012, Adoue V et al., Immunity 2019). By blocking loci that could antagonize the induced phenotype, HP1-dependent chromatin mechanisms thus stabilize Th cell identity over the long term. Like all epigenetic regulation, however, it should be noted that these repressive mechanisms are stable and transmissible through the cell cycle, but also reversible. In particular, chronic exposure to type I and II IFNs can reprogram fully-committed Th2 cells towards a hybrid Th2/Th1 phenotype (Hezagy AN et al., Immunity 2010). Epigenetic regulations are therefore rheostats that adjust the stability of gene expression programs without ever completely locking them, even in the case of heterochromatin-dependent repression mechanisms.

Using HP1 α/γ KO naïve CD4⁺ T cells, we show that HP1 α and HP1 γ differentially and non-redundantly regulate the crosstalk between Tconv and Treg. Inactivating HP1 α expression at the genetic level, to reduce CAR-T cell sensitivity to immunosuppressive signals for example, should therefore not be functionally compensated by other chromatin proteins. Whether chronic exposure to suppressive environmental cues will nevertheless succeed in repressing HP1 α KO T cell functions remains to be studied. Unfortunately, we were unable to answer this question with our experimental system. Indeed, as we previously published, T-cell mediated graft rejection occurs within days following T cell transfer, and tolerance to bone marrow allografts, once established, relies in the long term on central (*i.e.* intrathymic) rather than peripheral T cell tolerance mechanisms. (Pasquet L et al., Blood 2013). To deal with this point, we are currently setting up a model of transplanted tumor to analyze the anti-tumor response mediated by wild-type or HP1 α -deficient endogenous or genetically-engineered CD4⁺ and CD8⁺ T cells.

3) The authors nicely describe HP1 α and HP1 γ as paralogs that have the same structure and functional domains and undergo homo- or heterodimerization with binding partners that contain a PXVXL motif. Do the authors have any further insight into the sequence differences or variations in transcription factor interaction that could account for the opposite role of HP1 α and HP1 γ ?

Without experimental system that fully recapitulates *in vitro* the phenotypes observed *in vivo*, we cannot address this question experimentally. The number of allospecific T cells purified from transplanted mice was far too low to allow us to analyze the heterogeneity of HP1 α/γ complexes by proteomics. We therefore addressed this question using *in silico* approaches. We first intersected a comprehensive list of transcription factors and chromatin-associated molecules (n=1469) expressed in naive and differentiated CD4⁺ T cells (Stubington MJT et al., Biol Direct 2015) with the lists of HP1 α (n=214) and HP1 γ partners (n=194) referenced in the STRING database (<https://string-db.org/>).

This strategy enabled us to identify 57 et 53 potential partners for HP1 α and HP1 γ , respectively. As expected, the lysine methyltransferases Suv39h1, Suv39h2 and SETDB1, the co-transcriptional regulators TRIM24 and TRIM28 and the histone deacetylases (HDAC) 1 and 2 were present in both lists. We filtered out partners common to both variants to identify those specific to HP1 α (n=27) and HP1 γ (n=25). Among them, JunD and STAT3 were predicted as specific binding partners for HP1 γ (Cbx3) and HP1 α (Cbx5), respectively (see Figure above). Interestingly, STAT3 is activated downstream of numerous immunoregulatory molecules. They include receptors for IL-10, IL-35 and IL-27. Downstream of these receptors, HP1 α could be targeted on *cis*-regulatory elements by STAT-3 to repress Th1 genes. With regard to HP1 γ , its interaction with JunD, a subunit of the AP-1 complex, may help limit the trans-activating effect of this transcription factor on genes encoding immune checkpoints. AP-1 has been identified as a transcriptional regulator of *Pdcd1* downstream of the TCR. (Chamoto K, Nat Rev Immunol 2023). These *in silico* predictions

allowed us to identify distinct molecular partners for the two HP1 variants that could explain our *in vivo* phenotypes. Their experimental validation will be the subject of future work.

Reviewer #2 (Remarks to the Author):

The authors have examined the role of three related chromodomain proteins that can mediate epigenetic changes in CD4⁺ and CD8⁺ T lymphocytes. This report claims a specific effect of two of these proteins HP1a and HP1g on the function of T regulatory cells in vivo in a complex model of Treg mediated acceptance of allogeneic bone marrow transplantation. The results suggest that HP1a controls the capacity of Treg to mediate suppression of effector cytokine production so that in its absence cytokine production (Th1 and TH17) by T effector cells is not suppressed and allogeneic bone marrow is rejected. Conversely, Th1g has the opposite effect and in its absence results in an effector population that is easier to suppress with enhanced acceptance of an allogeneic transplant. This is a very novel concept, but there are numerous technical and theoretical problems with this report that weaken the conclusions drawn by the authors.

1. The mouse strains used as the effector populations are not genetically identical. Wild type B6 CD4⁺ T cells are used as the control populations while the three knock out strains are maintained on a mixed 129/B6 background. This may introduce unknown variations in the graft rejection studies.

We thank the reviewer for bringing up this important point. As indicated in the *Mice* section of the methods, we systematically “used and compared sex-matched 6- to 12-weeks-old wild-type and mutant littermates in all experiments”. Control and mutant CD4⁺ T cells used in the same experiment were therefore systematically isolated from wild-type and mutant littermates, respectively. We apologize for not being clearer in the first version of the manuscript, but we never used B6 mice as a source of control T cells. We have now modified the “Mice” and “Isolation of T-cell subsets” sections of the Methods to make them more explicit.

Furthermore, one of the knockout strains is a global knockout while the other two (HP1b and HP1c) are T cell conditional knockouts. Differential effects on T cell development may result in these different strains.

In this study, we used a bone marrow allograft model in which graft rejection was mediated by adoptively-transferred naive CD4⁺ T cells. The HP1-dependent chromatin mechanisms that we have critically implicated in the Tconv response to immunosuppressive signals are therefore intrinsic to these cells, the functionality of wild-type and mutant cells having been tested in the same environment.

We agree with the reviewer, however, that effects extrinsic and even intrinsic to T cells could have impacted their development and/or homeostasis in (constitutive) KO mice, and could therefore potentially be at the origin of the observed phenotypes. To address this issue, we first analyzed the intrathymic development of these cells by flow cytometry. We observed no differences in the representation of the four main subpopulations of thymocytes (Figures S1D-S1E and S4F-S4G) as well as in the thymic output of Treg (Figures S1F and S4H). When we analyzed the T cell compartment in peripheral lymphoid organs, we again failed to detect significant differences between control and (constitutive) KO strains. The proportion of CD4 and CD8 T cells (Figures S1G-S1H and S4I-S4J), the frequency of Treg within the CD4 compartment (Figures S1I and S4K) and the frequency of naïve, effector and memory CD4⁺ and CD8⁺ T cells (**new Figures S1J-S1M and S4L-S4O**) were similar in WT and mutant mice. We next characterized in more details the programming of the naive CD4⁺ T cells used in our experimental system but our differential analysis by RNA-seq failed to identify significant differences between the different genotypes (**new Figures S1P and S4R**). When we compared the transcriptome of naive CD4⁺ T cells isolated from HP1 α constitutive KO mice or control littermates, we only identified 12 differentially-expressed genes (Figure 3A and **new Figure S1P**) with no known significant role in T cells. Importantly, the global chromatin landscapes of HP1 α KO and control naive CD4⁺ T cells were also very similar (Figure 3E). Altogether, these observations strongly support that T cell development was not biased by HP1-dependent T cell-extrinsic mechanisms in the constitutive KO strains.

2. While the paper and previous studies by this group have examined the effects of Treg on the control of allogeneic graft rejection, the studies in the present report also involve the response to the superantigen mmtc-7 so this is not a pure allo response. More importantly, in many of the figures the authors restrict their analysis to the T cell subset (Vb6) which recognizes the superantigen. It would have been preferable to illustrate the global anti-allo response. In particular, the capacity of superantigens to induce pTreg or so called anergic cells may not be representative of the typical response to alloantigens.

In our bone marrow allograft model, recipient mice are preconditioned by lethal irradiation. In response to the induced lymphopenia, CD4⁺ T cells undergo strong proliferation and expansion. It is therefore impossible, on the basis of activation markers such as CD25, CD44 or CD69, to discriminate allospecific T cells from those proliferating in response to self-recognition events and IL-7. To limit the impact of lymphopenia-induced proliferation in our analyses, we therefore restricted our study to TCR Vβ6⁺ cells. To analyze the global conversion of allospecific T cells into anergic or regulatory cells, one option would have been to amplify these cells *ex vivo* with allogeneic antigen-presenting cells. We chose not to use this strategy because it would have generated major biases in the representation of the different subtypes within the CD4⁺ T cell compartment. Indeed, effector, regulatory, anergic and exhausted T cells have very different proliferative responses to TCR engagement. For a more global view of the impact of HP1γ on the alloresponse, we analyzed the percentage of anergic and regulatory cells among total CD4⁺ T cells. As observed in analyses restricted to TCR Vβ6⁺ cells, the results obtained show that mutant cells convert more efficiently into anergic (ratio 1:1) and regulatory (ratio 1:5) cells than wild-type cells (see Figure below). We hope these results will reassure the reviewer as to the relevance of our analyses.

3. The major problem with the paper is the characterization of the effector cell populations used to induce graft rejection. While the authors claim that CD4 T cell differentiation in the steady state is completely normal in the knock out strains, their analysis is rather superficial. One potential explanation is that the effector cells from the HP1a^{-/-} mice may contain a higher proportion of memory or primed T cells and are harder to suppress, while the effector cells from the HP1g mice have fewer memory cells and are easier to suppress. The authors claim to have used naive T cells in all their studies, but they need to document that the percentages of these populations are equivalent to the wild type control. T cell suppression in the graft rejection model is subtle and the authors claim in previous papers can only be seen with Treg populations that have been expanded *in vitro* as in the present paper.

We acknowledge the reviewer for his/her constructive comment. While one of the main functions of Treg is to suppress T cell priming, they can also inhibit effector and memory (CD4⁺) T cells (Suvas S. et al., J Exp Med. 2003; Levings MK et al., J Exp Med 2001). However, as the reviewer rightly pointed out, it has been shown that allograft rejection mediated by memory T cells can be resistant to regulation (Yang J et al., PNAS 2007). Consequently, an increased or reduced frequency of memory cells in the KO

populations could indeed explain the differences in susceptibility of control and mutant cells to Treg-mediated suppression. To test whether our observations could be due to this bias, we have carried out a large set of additional analyses and experiments. They showed that:

- The frequency of naïve, effector-memory and central-memory CD4⁺ and CD8⁺ T cells was similar in control and mutant mice (**new Figures S1J-S1M and S4L-S4O**).
- The purity of control and mutant Tconv suspensions in naïve CD4⁺ T cells was comparable and systematically greater than 95% (**new Figures S1Q and S4S**).
- Freshly purified control and mutant naïve CD4⁺ T cells (defined as CD4⁺CD25⁻CD44^{low}CD62^{high}) did not differentially express memory and/or effector markers (**new Figures S1P and S4R**).

Altogether, these data strongly support that our findings did not result from differential contamination of control and mutant naïve CD4⁺ T cell populations by effector or memory T cells.

4. The claim that the HP1a effector cells are only resistant to suppression of cytokine production but not expansion in vivo is somewhat unusual as most studies of Treg suppression in vivo show marked suppression of effector cell expansion in addition to cytokine production.

While most studies document concomitant inhibition of T cell proliferation and effector functions by Treg, the uncoupling we observed is not unprecedented. In a seminal paper, Chen ML and colleagues (Chen ML et al PNAS 2004) showed early on that Treg could inhibit the cytotoxic activity of tumor-specific CD8⁺ T cells without affecting their activation, proliferation, homing and cytokine production. Since then, similar observations have been reported in other studies (DiPaolo RJ et al., J Immunol 2005; Mempel TR et al., Immunity 2006 ; Dorothy KS et al, PNAS 2011). In a murine model of gastritis, Ethan M. Shevach's laboratory reported that Treg repressed the differentiation of autoreactive Tconv into Th1 cells without inhibiting their expansion into draining lymph nodes (DiPaolo RJ et al., J Immunol 2005). In their system, as in ours, IFN γ production by Tconv was therefore inhibited by Treg without affecting their accumulation. These works are now discussed in the discussion of the manuscript.

5. The fate of the transferred Treg in these different combinations of effector cell mice is not examined.

To determine the putative impact of HP1 α -deficiency in Tconv on Treg programming, we analyzed the transcriptome of Treg isolated from mice grafted with the BM mixture and injected with Treg and control or mutant Tconv. As shown in the **new Figure 2E**, differential transcriptome analysis of Treg exposed to KO and WT Tconv did not reveal any biologically significant differences. HP1 α deletion in effector T cells therefore has no extrinsic effect on Treg.

6. It is also surprising that the RNA-seq studies on the HP1 deficient strains did not reveal any abnormalities, but the authors limited their studies to naïve cells and not memory populations generated in vivo.

We have now generated a large set of additional data and analyses that largely confirmed that HP1 α - or HP1 γ -deficiency had no major impact on the T cell subpopulations in the steady-state. They showed that:

- The frequency of naïve, effector-memory and central-memory CD4⁺ and CD8⁺ T cells was similar in control and mutant mice (**new Figures S1J-S1M and S4L-S4O**).
- The transcriptome of control and mutant memory CD4⁺ T cells (*i.e.* CD4⁺CD25⁻CD62L^{low}CD44^{high}) showed only minor differences (**new Figures S1O and S4P**). Only very few genes were differentially expressed between wild-type and KO memory T cells and gene ontology analyses of these genes failed to retrieve any functionally relevant biological process.
- The repertoire of cytokines produced by control and mutant memory CD4⁺ T cells following acute *ex vivo* restimulation was similar (**new Figures S1N and S4P**).

Why are abnormalities in cytokine gene expression only seen after exposure to Treg in vivo?

Our *in vitro* and *in vivo* data show that HP1-deficiency has no major impact on Th1 and Th17 cell priming. These results suggest that HP1 α and HP1 γ do not regulate the epigenetic status of effector genes in naive CD4⁺ T cells. In support of this hypothesis, our ChIP-seq data did not reveal enrichment for the repressive mark H3K9me3 at the *cis*-regulatory elements that control *Ifng* and *Il17a* expression (see Figure below). This observation could explain why the genes encoding the signature cytokines of the Th1 and Th17 lineages are mobilized and expressed normally in response to activating signals.

Our data suggest that abnormalities in cytokine production are only revealed in the presence of Treg because HP1 α repress the effector program in Tconv only when mobilized by Treg-dependent suppressive signals, and because HP1 γ limits Th cell sensitivity to Treg-mediated suppression by repressing the TCR-induced expression of immune checkpoints.

7. No data is presented to support the claim that the anergic cells generated from HP1 $g^{-/-}$ effector cells are in fact exhausted. Since they express Vb6, they could easily be restimulated with MMTv7 expressing cells. 8. While PD-1 is a marker of exhausted cells, it is also a marker of recently activated cells. The exhausted phenotype should be tested again with MMTv stimulation.

To address these very interesting questions, we analyzed cytokine production (following acute restimulation with PMA and Ionomycin) and immune checkpoint expression by TCR V β 6⁺ Tconv from mice injected with HP1 γ KO Tconv only, or by anergic (A) or non-anergic (N-A) TCR V β 6⁺ Tconv from mice injected with HP1 γ KO Tconv and Treg (**new Figures 5E-5H**). Interestingly, in addition to expressing CD73 and FR4, the anergic cells fail to produce IFN- γ , IL-2 and TNF (**new Figure 5E**) and a large fraction of them co-express the immune checkpoints PD-1, LAG-3 and TIGIT (**new Figures 5F-5H**). These results, combined with our NGS analyses (**Figures 6D-6G**) demonstrate that the anergic cells generated from HP1 $g^{-/-}$ effector cells are in fact exhausted.

9. One more theoretical issue is how do the Treg mediate suppression--directly by contact with the T effector cells or indirectly via acting on APC (as is widely believed by some in the field)?

What triggers the activity of members of the HP1 family? TCR signals? Costimulatory signals?

In our bone marrow allograft model, we previously showed that Treg-mediated suppression critically involves soluble mediators (Joffre et al., Nat Med 2008). Using Tconv expressing a dominant-negative TGF- β RII mutant, we demonstrated that allograft protection requires effector T cells responsiveness to TGF- β .

In vitro priming of wild-type and HP1 α KO T cells in the presence of TGF- β did not, however, recapitulate the phenotype observed *in vivo* (data not shown). As we previously also ruled out a role for Treg production of IL-10 in the induction of allograft tolerance, we tested whether Treg-mediated suppression of allospecific Tconv could involve cell contacts. The hypothesis of a role for inhibitory receptors was reinforced by the importance of HP1 γ in regulating the TCR-induced expression of immune checkpoints (**Figures 6H-6K**), and by the fact that HP1 γ KO Tconv are more efficiently repressed by Treg than wild-type Tconv. To test whether immune checkpoints might also be involved in the suppression of the alloresponse by Treg, we treated mice previously injected with Tconv and

Treg with blocking antibodies specific for PD-1 and/or LAG-3. Unfortunately, regardless of the clones of anti-PD1 (J43 or 29F.1A12) and anti-LAG3 (C9B7W) antibodies used, and whatever the amounts and administration protocol tested, the injection of the immune checkpoint inhibitors systematically led to Tconv and Treg depletion. We haven't been able to explain this result, but to counteract the problem we are now setting up a model of transplanted tumors in non-lymphopenic hosts to address the same question. We hope that this new study will enable us to verify whether inhibitory receptor-mediated suppression involves HP1 α and, at the same time, to assess the relevance of our discovery in cancer therapy.

Reviewer #3 (Remarks to the Author):

The manuscript by Joffre and colleagues focuses on the well-known phenomenon that the activation of effector T cells can be controlled by Treg. Although several mechanisms are known by which Treg exert the suppression, knowledge of signaling and consequences in the target cells is incomplete. The submitted manuscript describes a link between the Treg-mediated suppression and the heterochromatin proteins alpha and gamma. Using a mouse model of bone marrow transplantation, the authors show that HP1a-deficient cells fail to repress their effector mechanisms and that HP1g-deficient cells are more efficient in suppressing effector T cells. In addition, the authors prepared T cells from both knockout scenarios subjected them to ATAC-seq and RNA-seq and identified key pathways/genes for the observed effects.

Overall, I recommend the manuscript for publication as it is sufficiently novel and of interest to a wider group of scientists. However, a few points need to be clarified and improved before the paper can be finally approved.

Major points

The statistical method in Figure 1C, 2B-D, 3G, 4B,D, 5B-F, 6B, I, K is unclear.

All the statistical tests used and the values represented by asterisks are now indicated in the legends of the main and supplementary figures. We apologize for these omissions.

The effect shown in Figure 2E was interpreted by an unaltered Treg-induced contraction. The gating of the dot plot and the explanation is not completely clear to me. Could the authors exclude that the CD45.2+ T cell reduction is just a consequence of filling the T cell niche in the presence of another population in the spleen? Is it competition or Treg-induced contraction in the used model? Please provide more data for your interpretation.

Thank you for your comment which enabled us to detect that our gating strategy was biased by variations in the relative proportions of syngeneic vs allogeneic hematopoietic cells depending on the experimental conditions. We reanalyzed our data to determine the % of Tconv, defined as CD4⁺CD45.2⁺H-2Kd⁻Thy1.1⁺, among total spleen cells (rather than among H-2Kd⁻ cells as we had initially done). Interestingly, this analysis showed that the co-injection of Treg markedly decreased the frequency of allospecific (*i.e.*, Vβ6⁺) T cells (**new Figure 2H**) without inducing an overall contraction of the CD4⁺ T cell repertoire (**new Figures 2F-2G**). This specificity of action suggests that the contraction observed is mainly induced by the immunosuppressive activity of Treg rather than by their competition with Tconv to fill the T cell niche.

The authors identified 54 genes that were more accessible and expressed in HP1a Ko cells compared to WT cells. However, these genes are not listed despite the presentation of pathway analyses in Figure 3. It would be beneficial to include a list of the genes with associated values or as a heatmap (might be reduced to top hits). The authors should consider moving some of the moderately informative data, such as Figure 3I, to the supplement. The same applies to the similar analysis of the HP1g Ko cells. Figure 6E lists only a few genes, with no information about the remaining genes and their values. The genes from both experiments should appear in the discussion.

Thank you for your comment. We have now added two tables (**new Supplementary Tables 1 and 2**) in which the name of the genes, their expression level in WT and KO cells and the p-value are provided.

Treg were expanded before transfer. Did the authors purify the Treg after culture? What was the purity of the transferred Treg? This should be mentioned in Material and Methods. Please provide a quality check of the Treg by flow cytometry in the Supplement.

For the *in vitro* experiments, mouse Treg were purified by positive selection from the spleen of Foxp3-Thy1.1 mice. The procedure included two consecutive selections on MS columns, making it possible to

obtain a cell suspension systematically enriched to over 95% in CD4⁺Thy1.1⁺ cells. For the *in vivo* experiments, the resulting cell suspension underwent an additional selection step to culture a virtually pure Treg population. Treg, defined as CD4⁺Thy1.1⁺CD25⁺, were FACS-sorted on a FACSaria Sorp or Fusion. At the end of the culture, the Treg population obtained was routinely more than 95% pure (**new Figure S1R**).

Minor points

Please introduce TCR Vb6 earlier. Explanation is in Line 189-190, but appears already at the beginning of this result paragraph.

TCR Vb6 is now introduced earlier, when the data presented in Figures 2A-B are described. Thank you for the suggestion.

Figure 6A: Title for color range is missing.

The title has now been added. Sorry for the oversight.

Figure S4 I,K: Please add a title for the organ.

As requested, we have now specified the organ analyzed for panels S4F to S4R (as well as S1D to S1P).

Figure 6A: Which comparison was used to define the clusters? Tconv group? Please ensure that it is clearly stated in the text.

As indicated in the section "Correlated modules definition" of the Methods, *the analyses were run on genes overexpressed in HP1 γ KO Tconv exposed to Treg compared with wild-type Tconv exposed to Treg*. It is now clearly stated in the text.

The wording "..., or not, ..." in the figure legends is a bit unusual and difficult to understand. It is recommend to revise the legends and use easy-to-understand descriptions.

The wording "..., or not, ..." has been removed and the figure legends have been revised as requested.

Please also revise the methods:

1. Some paragraphs can be shortened.

As requested, any information that was not absolutely necessary for understanding and reproducing the experiments was removed.

2. Provide exact information for antibody fluorochromes (e.g. Line 544).

The coupling of all antibodies used in flow cytometry is now indicated.

3. Were the Treg isolated by two different methods? MACS or MACS and FACS? Please, indicate the method used for each type of experiment.

The Treg were first enriched by positive selection from the spleen of *Foxp3*-Thy1.1 mice and then FACS-sorted based on CD4, CD25 and Thy1.1 expression. We now modified the first paragraph of the section "Isolation of mouse and human T-cell subsets" to express the procedure more clearly.

4. Line 706: Is Nextflowewel a typo?

It is indeed a typo. It should have read 'NextFlow'. Sorry, the error has now been corrected in the manuscript.

Reviewer #4 (Remarks to the Author):

This study meticulously investigates the role of HP1 α and HP1 γ in regulating T cell responses, particularly focusing on their impact on the interaction between Tconv and Treg. The authors employed a well-characterised mouse model of bone marrow transplantation to elucidate the molecular mechanisms underlying the crosstalk between Tconv and Treg, revealing novel insights into the heterochromatin-dependent regulation of T cell sensitivity to immunosuppressive signals.

The findings demonstrate that Tconv deficient in HP1 α exhibit reduced sensitivity to Treg-mediated suppression, leading to inefficient control of allogeneic responses. This observation is further corroborated by the resistance of HP1 α -deficient Th1 and Th17 cells to Treg-mediated inhibition, suggesting a critical role of HP1 α in modulating T cell responses through chromatin remodelling. The comprehensive analysis, including RNA-seq and ATAC-seq, provides compelling evidence that HP1 α deficiency leads to an upregulation of genes associated with T cell activation and Th17 cell-dependent immunity, thus highlighting the gene networks regulated by HP1 α in the context of Treg-mediated suppression.

In contrast, the study also delves into the role of HP1 γ , presenting intriguing findings that HP1 γ deficiency enhances Treg's ability to suppress allogeneic T-cell responses. This effect is attributed to the modulation of a gene network associated with T cell exhaustion and anergy by HP1 γ , as evidenced by the differential expression and accessibility of genes involved in immune checkpoints and immunosuppressive cytokine production in HP1 γ -deficient cells.

The experiments are well-designed and include enough details. However, there are some questions and comments as below:

1. Model System Relevance: The study predominantly relies on a mouse model of bone marrow transplantation to elucidate the role of HP1 α and HP1 γ . While this model is invaluable for dissecting molecular mechanisms, the direct translatability of findings to human biology and disease contexts may be limited. Differences between mouse and human immune systems could affect the applicability of these results in clinical settings. **2. Lack of Human Data:** The absence of validation in human T cells or patient samples limits the study's immediate relevance to human health and disease. Incorporating experiments using human cells could help verify the conservation of observed mechanisms across species and enhance the study's clinical significance. **Did the authors test any samples from patients with autoimmunity or cancer, for instance?**

Thank you for raising these very important points. We used two complementary approaches to begin testing whether our observations could be relevant in the clinic:

- To assess whether targeting HP1 α could be of interest in cancer patients, we first compared the transcriptomic signature induced by HP1 α -deletion in murine CD4⁺ T cells to the transcriptome of tumor-infiltrating T lymphocytes (TILs) isolated from patients with advanced melanoma (**new Figures 3K-3N**). Interestingly, we observed a significant enrichment of genes downregulated in HP1 α -deficient cells among those that define human exhausted T cell identity (**Figure 3N**). These data suggest that inhibiting HP1 α in tumor-specific human CD4⁺ T cells could, as it does in allospecific murine Tconv (**Figures 3C and 3D**), repress the establishment of the exhaustion program and thus promote a protective immune response.
- To test whether HP1 γ -deficiency could sensitize human CD4⁺ T cells to immunosuppressive signals, we set up a mouse model of xenogeneic graft-versus-host-disease (xGvHD) in which immunodeficient NSG hosts are injected with human naive CD4⁺ T cells alone or in the presence of human Treg (**new Figure 7A**). In this system, and in the absence of suppressive cells, a strong xenogeneic Th1 response developed within weeks of injection (**new Figures 7D-7H**). At a Treg:Tconv ratio 2:1, the co-injection of suppressor cells does not significantly repress

xenospecific wild-type T cells. In contrast, under the same conditions, Treg effectively repress the Th1 response mediated by human naive CD4⁺ T cells in which HP1 γ -expression had previously been inactivated by CRISPR-Cas9 (**new Figures 7D-7H**). Interestingly, some rare but detectable mutant cells also produce the anti-inflammatory cytokine IL-10 and express the transcription factor Foxp3 in the presence of suppressor cells (**Figures 7I and 7J**). Taken together, these *in vivo* data suggest that HP1 γ regulates the susceptibility of human CD4⁺ T cells to immunosuppressive signals, and that targeting this molecule could be clinically relevant for repressing pathogenic T cell responses in transplanted individuals or in patients suffering from autoimmunity.

These data therefore support the idea that the HP1-dependent chromatin mechanisms we identified in mice could also be at work in human T cells.

3. Long-term Effects and Immune Tolerance: The study focuses on relatively short-term outcomes following bone marrow transplantation and T cell transfer. The long-term effects of HP1 α and HP1 γ deficiency on immune tolerance, autoimmunity, and memory T cell responses are not explored. These aspects are crucial for understanding the full implications of HP1 α/γ regulation in immune homeostasis and disease. This question has clinical implications, for instance, the potential role of the identified mechanisms in longer-term conditions such as chronic GvHD.

We thank the reviewer for his/her very interesting question. HP1-dependent gene silencing mechanisms have been described as stable and inheritable through multiple cell divisions. Homodimerisation of HP1s *via* their chromoshadow domain results in the formation of a dense protein network that locks up chromatin in a highly compacted state (Canzio D et al., Nature 2013). In differentiating Th cells, we and others already showed that HP1s and partners (*i.e.* SETDB1, SUV39H1) are critically involved in the repression of genes associated with alternative fates (Allan RS et al., Nature 2012, Adoue V et al., Immunity 2019). By blocking loci that could antagonize the induced phenotype, HP1-dependent chromatin mechanisms thus stabilize Th cell identity over the long term. Like all epigenetic regulation, however, it should be noted that these repressive mechanisms are stable and transmissible through the cell cycle, but also reversible. In particular, chronic exposure to type I and II IFNs can reprogram fully-committed Th2 cells towards a hybrid Th2/Th1 phenotype (Hezagy AN et al., Immunity 2010). Epigenetic regulations are therefore rheostats that adjust the stability of gene expression programs without ever completely locking them, even in the case of heterochromatin-dependent repression mechanisms.

Using HP1 α/γ KO naïve CD4⁺ T cells, we show that HP1 α and HP1 γ differentially and non-redundantly regulate the crosstalk between Tconv and Treg. Inactivating HP1 α expression at the genetic level, to reduce CAR-T cell sensitivity to immunosuppressive signals for example, should therefore not be functionally compensated by other chromatin proteins. Whether chronic exposure to suppressive environmental cues will nevertheless succeed in repressing HP1 α KO T cell functions remains to be studied. Unfortunately, we were unable to answer this question with our experimental system. Indeed, as we previously published, T-cell mediated graft rejection occurs within days following T cell transfer, and tolerance to bone marrow allografts, once established, relies in the long term on central (*i.e.* intrathymic) rather than peripheral T cell tolerance mechanisms. (Pasquet L et al., Blood 2013). To deal with this point, we are currently setting up a model of transplanted tumor to analyze the anti-tumor response mediated by wild-type or HP1 α -deficient endogenous or genetically-engineered CD4⁺ and CD8⁺ T cells.

4. Mechanistic Insights into HP1 α/γ Function: While the study provides significant insights into the consequences of HP1 α and HP1 γ deficiency on T cell function and regulation, the precise molecular mechanisms through which these proteins exert their effects remain partially elucidated. Also, a crucial aspect, for instance, is whether this change of function is cell contact-dependent. Can you see the same effect in a trans-well culture?

In our bone marrow allograft model, we previously showed that Treg-mediated suppression critically involves soluble mediators (Joffre et al., Nat Med 2008). Using Tconv expressing a dominant-negative TGF- β RII mutant, we demonstrated that allograft protection requires effector T cells responsiveness to TGF- β .

In vitro priming of wild-type and HP1 α KO T cells in the presence of TGF- β did not, however, recapitulate the phenotype observed *in vivo* (data not shown). As we previously also ruled out a role for Treg production of IL-10 in the induction of allograft tolerance, we tested whether Treg-mediated suppression of allospecific Tconv could involve cell contacts. The hypothesis of a role for inhibitory molecules was reinforced by the importance of HP1 γ in regulating the TCR-induced expression of immune checkpoints, and by the fact that HP1 γ KO Tconv are more efficiently repressed by Treg than wild-type Tconv. To test whether immune checkpoints might also be involved in the suppression of the alloresponse by Treg, we treated mice previously injected with Tconv and Treg with blocking antibodies specific for PD-1 and/or LAG-3. Unfortunately, regardless of the clones of anti-PD1 (J43 or 29F.1A12) and anti-LAG3 (C9B7W) antibodies used, and whatever the amounts and administration protocol tested, the injection of the immune checkpoint inhibitors systematically led to Tconv and Treg depletion. We haven't been able to explain this result.

A few specific questions:

-Page 7: line 171: Authors say, “in the absence of Treg...” I am not sure why the presence or absence of Treg is relevant to the phenocopy of wild-type cells? Could you please explain?

We just wrote ‘in the absence of Treg’ to make it clear that we had only injected Tconv and that in these conditions HP1 α -deficient cells behaved like wild-type cells. We have now removed this detail to avoid confusion.

-Page 9, line 270: “When cultured under Th1- or Th17-polarizing conditions, wild-type and HP1 γ -deficient T cells also expressed similar levels of the lineage-determining master regulators T-bet and ROR γ t” Did you do the same for HP1 α ?

Similar experiments have been performed using control and HP1 α -deficient cells (**Figures S2F-S2O**). As indicated in the manuscript “control and HP1-deficient cells showed similar activation and proliferation upon T cell receptor triggering (**Figures S2A-S2E**), and comparable Th1 and Th17 priming when exposed to lineage-specifying cytokines (**Figures S2F-S2O**)”.

-Page 11: Did you try to block PD-1 and LAG3 on HP1 γ -deficient T cells and compare the function?
Please, refer to point 3.

-Page 14: “Interestingly, our data suggest that inhibition of the SUV39H1-HP1 axis with compounds such as chaetocin, by boosting the expression of inhibitory coreceptors on the surface of T cells, could represent a synergistic treatment to these therapeutic strategies.” I agree, but did you try to test this in an in-vitro model?

We thank the reviewer for his/her constructive comment. We have now tested whether the upregulation of immune checkpoints observed in HP1 γ KO cells following TCR engagement could be reproduced by treating wild-type cells with chaetocin. Remarkably, we observed that acute inhibition of the lysine methyltransferase increased PD-1 and LAG-3 expression in activated wild-type CD4⁺ T cells in a manner similar to that observed in HP1 γ -deficient cells (**new Figures 6L-6N**).

Minor comment:

Several complex sentences take a few readings to understand and need simplification. For instance: “In contrast to their wild-type counterparts, HP1 α -deficient cells fail to efficiently repress their

effector mechanisms when exposed to Treg, while HP1 γ -deficient cells are more sensitive to Treg-mediated suppression.

We hope to have simplified any complex or confusing sentences.

Dear Referees,

We were delighted to read that you were convinced of the importance and robustness of our observations, and that you now support the publication of the article.

As requested by reviewer 3, we have added references to our statement '*We and others recently reported that transcriptional specificity is largely controlled by heterochromatin-dependent gene silencing in differentiating Th cells.*'

Yours sincerely,

Dr. Olivier JOFFRE